# Dynamic and intricate regulation by the Csr sRNAs in the Arctic *Pseudoalteromonas fuliginea*
Zedong Duan [1,2], Li Liao [1,2] ✉, Tingyi Lai[1,2], Ruyi Yang[2,3], Jin Zhang[2] & Bo Chen[2]

The Csr (Carbon Storage Regulator) system is pivotal in controlling various cellular functions in most bacteria, primarily through the CsrA protein and its antagonistic sRNAs. However, riboregulatory networks are less explored in non-model organisms, particularly those in extreme environments. In this study, we discovered two new sRNAs of the Csr system, Pf2 and Pf3, in the Arctic bacterium *Pseudoalteromonas fuliginea* BSW20308, along with the previously known Pf1. By studying the impact of these Pf sRNAs on CsrA targetomes and physiological processes, we found a significant influence on various cellular functions and a collective effect on the interaction dynamics between CsrA and RNAs. Furthermore, we identified additional sRNAs that can interact with CsrA and mRNAs. Overall, our results emphasize the growing influence of the Csr system on cellular physiology through intricate sRNA regulation of CsrA, revealing riboregulatory network complexity and significance in non-model organisms.

The carbon storage regulator (Csr) system, present in ~75% of bacterial species and regulating over 20% of all mRNAs in *E. coli*, is a widely occurring and pivotal global regulatory system[1–3]. The Csr system mainly consists of a conserved RNA-binding protein (RBP), i.e., CsrA or RsmA, and noncoding small RNAs (sRNAs) that bind to the protein competitively[4]. CsrA, initially identified as a carbon source metabolism regulator in *E. coli* in 1993, is a well-known globally acting RBP that functions at the post-transcriptional level[5]. Non-coding sRNAs, such as CsrB and CsrC, bind to CsrA through multiple high affinity binding sites, forming complexes that sequester CsrA, thereby prevent its interaction with lower affinity mRNA targets[6]. The Csr system plays critical roles in the global regulation of gene expression in various processes, including carbon metabolism, cell motility, biofilm formation, quorum sensing, virulence, iron metabolism, and secondary metabolism[1,7,8]. Understanding of the Csr system can provide insights into complex riboregulation and its evolution.

sRNAs are known to regulate gene expression post-transcriptionally, often by binding to mRNA or proteins[3,9–11]. The regulation of mRNA typically requires the assistance of Hfq, a global RNA chaperone that stabilizes sRNA-mRNA interactions and facilitates their base-pairing[9–11]. In contrast, sRNAs that regulate proteins often directly interact with the CsrA protein. CsrA antagonistic sRNAs, a key component of the Csr system, usually lack sequence conservation, making their identification by bioinformatics challenging. However, classical Csr sRNAs share repeated GGA motifs which are responsible for binding to CsrA. For example, the two Csr sRNAs in *E. coli*, CsrB and CsrC, contain 22 and 14 GGA motifs, respectively[12]. While at least one Csr sRNA is typically present in the Csr system, multiple Csr sRNAs have been identified in various species, including *E. coli* (CsrB and CsrC)[12], *Vibrio cholerae* (CsrB, CsrC, and CsrD)[13], *Vibrio tasmaniensis* (CsrB1, CsrB2, CsrB3, and CsrB4)[14], *Pseudomonas aeruginosa* (RsmW, RsmY, RsmZ, and RsmV)[15], and *Pseudomonas fluorescens* (RsmX, RsmY, and RsmZ)[16–18]. The presence of multiple Csr sRNAs appears functionally redundant, but the sRNAs may respond individually to distinct stimuli and vary in their ability to regulate CsrA to fine-tune the system[14,19]. Few studies have investigated the influence of varying numbers of Csr sRNAs on the collection of CsrA targets (the "targetome"), an essential aspect for understanding the significance of multiple Csr sRNAs and the robustness of the Csr system. It remains unclear whether the presence of different numbers of Csr sRNAs has any influence on the CsrA targetome.

Although the Csr system is widely distributed in bacteria[20], most of our knowledge regarding the Csr system has been gained from a limited number of microbes, such as *E. coli*[1], *Vibrio*[13], *Pseudomonas*[21], *Erwinia*[22], and *Salmonella*[9], with a predominant focus on model strains or pathogens[23]. However, the role of the Csr system in non-model strains, especially those

[1]Key Laboratory of Polar Ecosystem and Climate Change, Ministry of Education; Shanghai Key Laboratory of Polar Life and Environment Sciences; and School of Oceanography, Shanghai Jiao Tong University, Shanghai, China. [2]Key Laboratory for Polar Science, Ministry of Natural Resources, Polar Research Institute of China, Shanghai, China. [3]School of Health Science and Engineering, University of Shanghai for Science and Technology, Shanghai, China. ✉e-mail: liaoli@pric.org.cn

thriving in extreme environments, has not been well characterized. Additionally, the variability in the specific roles of the Csr system across different species suggests that its mechanisms of action might be diverse, underscoring the need for further research in non-model microorganisms[7]. Therefore, expanding the research scope to include non-model bacteria, extremophiles in particular, is essential for a comprehensive elucidation of the Csr system and for understanding its roles in diverse niches.

Previously, we discovered the first Csr sRNA, Pf1, in the marine bacterium *P. fuliginea* BSW20308, which possesses two chromosomes and is capable of thriving in the frigid Arctic Ocean[24,25]. Pf1 sRNA was identified as a new member of the CsrB family and was found to regulate a range of genes related to motility, carbon metabolism, and biofilm formation, among others, under temperature stress[24]. This highlights the significance of Csr sRNAs in adaptation to extreme environments. *Pseudoalteromonas*, renowned as a model genus for studying adaptation to cold environments, is globally distributed and particularly prevalent in polar oceans[24,26]. Therefore, studies on the Csr system in *P. fuliginea* BSW20308 broaden our understanding of this system within a different phylogenetic group and in extreme environments. Given that Csr sRNAs exist in multiple copies across various species[20], our study initially focused on identifying additional Csr sRNAs in *P. fuliginea*. Furthermore, we investigated the influence on the CsrA targetome in the absence of varying numbers of Csr sRNAs in *P. fuliginea*, to tackle the question of the presence of multiple Csr sRNAs in a more nuanced manner. Our research provides a comprehensive understanding of the regulatory role of Csr sRNAs within the CsrA-RNA network in *Pseudoalteromonas*, and offers profound insights into the global regulatory functions of the Csr system.

## Results

### Identification of two additional Csr sRNAs in *P. fuliginea* BSW20308

Given that CsrB family sRNAs usually consist of multiple copies across different species, we hypothesized that *P. fuliginea* BSW20308 may also possess multiple Csr sRNAs. Based on the presence of multiple GGA motifs, putative promoters, and rho-independent terminators, two additional putative Csr sRNAs, designated as Pf2 and Pf3, were identified (Fig. 1a). Like the previously identified Pf1 sRNA, both Pf2 and Pf3 sRNAs were in intergenic regions on the first chromosome. The Pf2 and Pf3 sequences contained 21 and 9 GGA motifs, fewer than the 25 GGA motifs found in Pf1, respectively (Fig. 1a, b). These findings, both in sequence and structure, align Pf2 and Pf3 with typical characteristics of Csr sRNAs.

The predicted promoters and the antagonistic function of Pf2 and Pf3 towards CsrA were further experimentally verified. Transcriptional fusions of Pf2 and Pf3 promoters with promoter-less GFP (green fluorescence protein) were constructed, and subsequent GFP fluorescence measurements demonstrated that both predicted promoters were functional (Fig. 1c). Binding of Pf2 and Pf3 to CsrA in vitro was verified by electrophoretic mobility shift assays (EMSA) (Fig. 1d). Furthermore, both Pf2 and Pf3 sRNAs were tested for their ability to bind CsrA heterologously in *E. coli* and regulate glycogen synthesis in vivo, following previous studies[12,24]. Glycogen synthesis, assessed by iodine staining, exhibited a notable increase with Pf2 and Pf3 overexpression compared to the empty plasmid control, paralleling the effects observed with CsrB in *E. coli* and Pf1 (Fig. 1e)[24]. Therefore, Pf2 and Pf3 were identified and verified as two additional Csr sRNAs in *P. fuliginea* BSW20308.

### Distribution of Pf sRNAs in *Pseudoalteromonas* and beyond

Having discovered three Pf sRNAs in *P. fuliginea* BSW20308, we postulated their broad distribution in the *Pseudoalteromonas* genus. To validate this, we scrutinized 672 *Pseudoalteromonas* genomes publicly accessible in the NCBI database. The comparative sequence analysis against Pf1, Pf2, and Pf3 revealed significant matches (with sequence identity ranging from 75.6% to 100% and *e*-values < 1e-5) in 252 (37.5%), 303 (45.1%), and 470 (69.9%) genomes, respectively (Supplementary Data 1). Notably, these homologous sequences were predominantly located in intergenic regions and exhibited a

high prevalence of GGA motifs, a hallmark of Csr sRNAs, reinforcing their potential classification as such. Particularly, 240 strains (35.7% of the total *Pseudoalteromonas* strains analyzed) possessed all three Pf sRNA-like sequences. This finding supports the widespread, yet not ubiquitous, presence of these three Csr sRNAs in the *Pseudoalteromonas* genus.

However, no significant matches for Pf1 and Pf2 sRNAs were found beyond the *Pseudoalteromonas* genus in the NCBI database. Nevertheless, for Pf3 sRNA, matches with over 60% sequence identity and 20–30% coverage (>100 bp) were found beyond *Pseudoalteromonas* in the NCBI database, including sequences predominantly from eukaryotes such as zebrafish (*Danio rerio*) and wasp (*Vespa crabro*), as well as the bacterial species *Thalassotalea* sp. LPB0316.

### Impact of Pf sRNAs deletion on growth and stress tolerance in *P. fuliginea* BSW20308

As previously reported in *V. cholerae*, *V. tasmaniensis*, and *P. fluorescens*, Csr sRNAs were redundant, and loss of one or several copies did not significantly affect phenotypes[13,14,18]. While multiple Pf sRNAs appear to be prevalent in *Pseudoalteromonas*, it remains uncertain whether same results would be observed in this genus. Here, we investigated the consequences of deleting one to three Csr sRNAs through the construction of ΔPf1, ΔPf12, ΔPf123 mutants derived from *P. fuliginea* BSW20308 (Supplementary Fig. 1). The primary objective of this study was to investigate the impact on the CsrA interactome and, subsequently, growth behavior in relation to varying numbers of Csr sRNAs. Consequently, we refrained from constructing all possible mutation combinations of Pf sRNAs. Our results showed that a single deletion had minimal influence on growth in 2216E media (Supplementary Fig. 2), consistent with previous studies on other species[27]. However, the influence became greater when more Pf sRNAs were deleted. Notably, the growth of the double (ΔPf12) and triple (ΔPf123) deletion mutants was severely compromised (Fig. 2a). Although the growth curves of ΔPf12 and ΔPf123 were nearly identical in standard 2216E media and in 2216E supplemented with 4% NaCl, the growth of ΔPf123 was notably inferior to that of ΔPf12 in 2216E media containing 4% sucrose. This finding generally aligns with the notion that these Pf sRNAs exhibit some degree of functional complementarity, although their performance may vary under specific conditions. This indicates that these sRNAs are functionally complementary to some extent.

Since Csr sRNAs were shown to regulate responses to salinity tolerance and osmotic stress in previous studies[24,28,29], strains were cultivated under various salinities and sucrose concentrations to evaluate the growth performance with varying numbers of Pf sRNAs deleted. In general, the salinity tolerance range did not significantly differ between the wild-type (WT) and sRNA mutant strains, with optimum growth observed at 2% (w/v) NaCl and no growth observed at 8% (w/v) NaCl for all strains (Supplementary Fig. 3). However, the deletion of Pf sRNAs did alter growth to some extent under all salinities tested, with an increased growth inhibition observed at a higher NaCl concentration (4%) (Fig. 2b). Nevertheless, under sucrose stress conditions, growth was severely inhibited in the double and triple deletion strains (ΔPf12, and ΔPf123) compared to the WT strain and ΔPf1 strain at both 2% and 4% (w/v) sucrose (Fig. 2c and Supplementary Fig. 3). While the Csr system displayed resilience against the deletion of a single Pf sRNA, its functionality was severely compromised by the elimination of all copies, particularly in the face of stresses such as osmotic stress induced by sucrose.

### RIP-seq comparison of the wild-type and mutant Pf sRNA strains

Given the varied growth observed in the mutations and the WT strain mentioned earlier, along with the fact that Csr sRNAs act as antagonists of CsrA, we hypothesized that the absence of Pf sRNAs could alter the CsrA targetome, i.e., the collection of targets that CsrA binds to and regulates. Therefore, we constructed 3×FLAG-tagged CsrA in the ΔPf1, ΔPf12, ΔPf123, and WT strains for RIP-Seq analysis. The expression of 3×FLAG-tagged CsrA is active in four strains, and the introduction of the FLAG-tag does not affect the *P. fuliginea* growth under the examined conditions (Supplementary Fig. 4). We performed co-immunoprecipitations (coIPs)

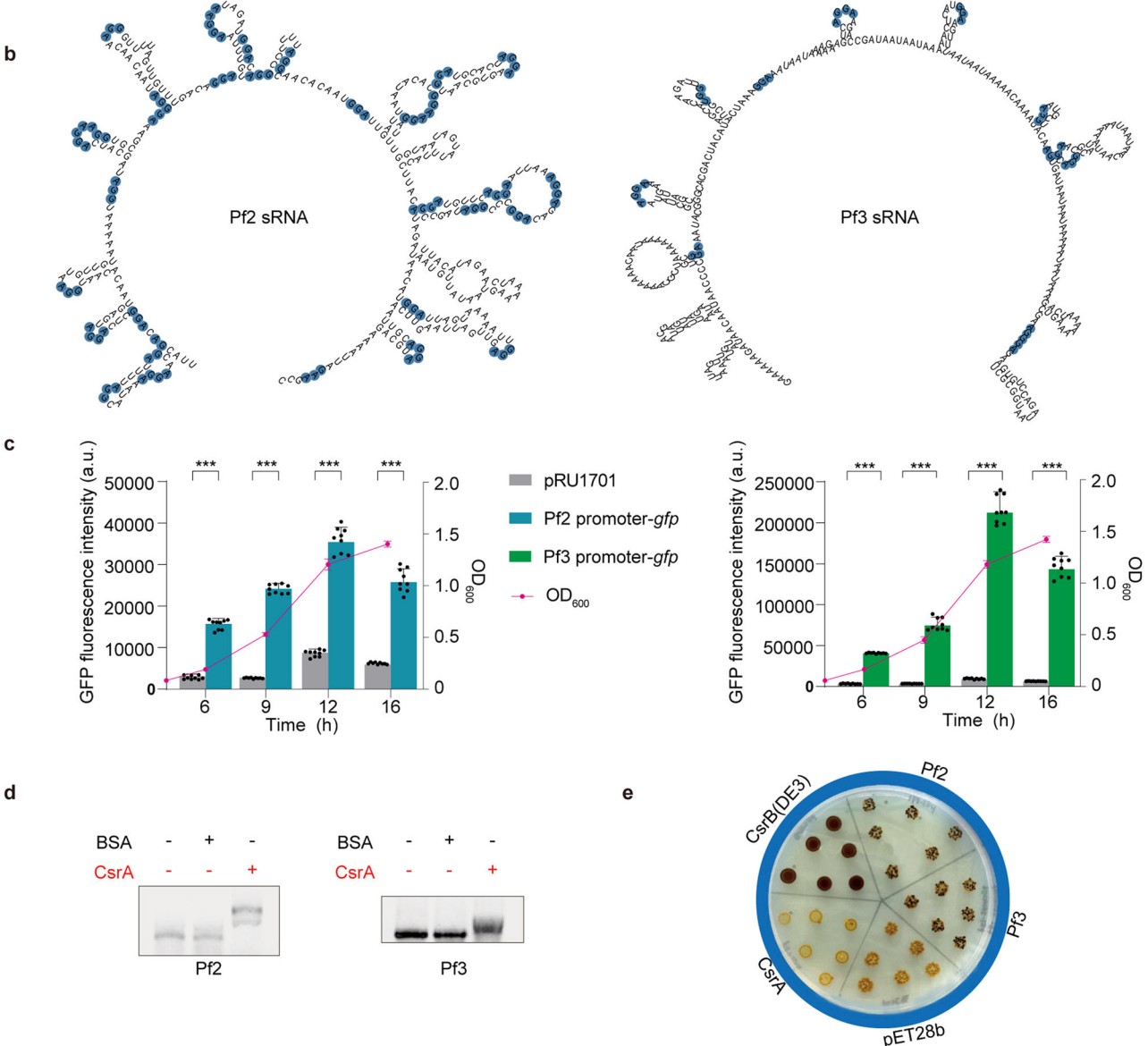

**Fig. 1 | Discovery and identification of Pf2 and Pf3 sRNAs. a** Nucleotide sequences and key features of Pf2 and Pf3 sRNAs. Pf2 is situated between the genes D172_RS02375 and D172_RS02370, while Pf3 lies between D172_RS16045 and D172_RS16040. Conserved upstream sequences of the CsrB family sRNAs are marked in orange. The promoters verified by initiating GFP expression are in blue. The "GGA" motifs are in pink. The rho-independent terminators are in red. **b** Secondary structures of Pf2 and Pf3 predicted by Mfold. The "GGA" motif sequences are displayed in blue. **c** Promoter activities of Pf2 and Pf3 during different growth phases. GFP fluorescence intensity (bars) driven by the Pf2 and Pf3 promoters was measured at 6 h, 9 h, 12 h, and 16 h after inoculation, with pRU1701 (empty vector) as a control. The $OD_{600}$ values (pink line) were recorded simultaneously to correlate promoter activities with bacterial growth phases. Plotted is the mean ± s.e.m ($n = 9$, ***$P < 0.001$ using Student's $t$-test). **d** EMSA analysis of 1 μM CsrA binding to Pf2 and Pf3, with 1 μM BSA as a negative control. **e** Role of Pf2 and Pf3 in glycogen synthesis assessed using iodine staining. Empty plasmid pET28b served as the blank control, pET28b-CsrA as the negative control, and pET28b-CsrB (DE3) as the positive control.

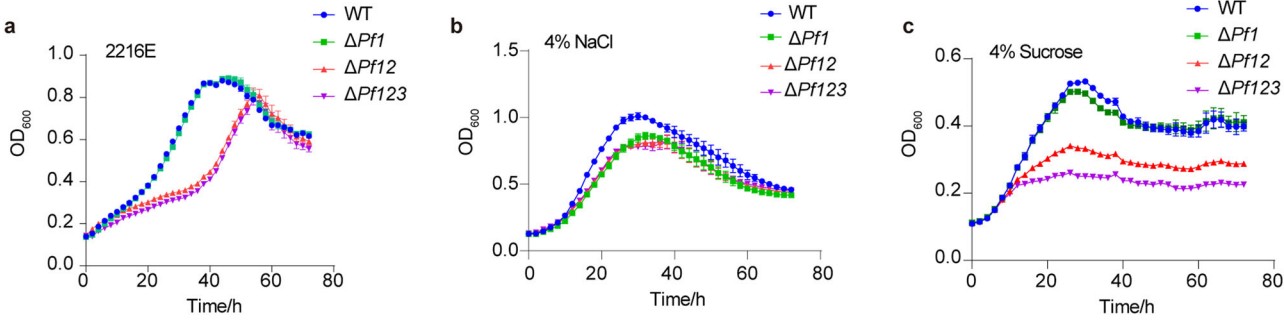

**Fig. 2 | Growth curves of the wild-type strain and Pf sRNA mutants under different conditions.** Including 2216E medium (**a**), 2216E medium with 4% NaCl (**b**) and 2216E medium with 4% sucrose (**c**). Additional growth curves are presented in Supplementary Figs. 2 and 3.

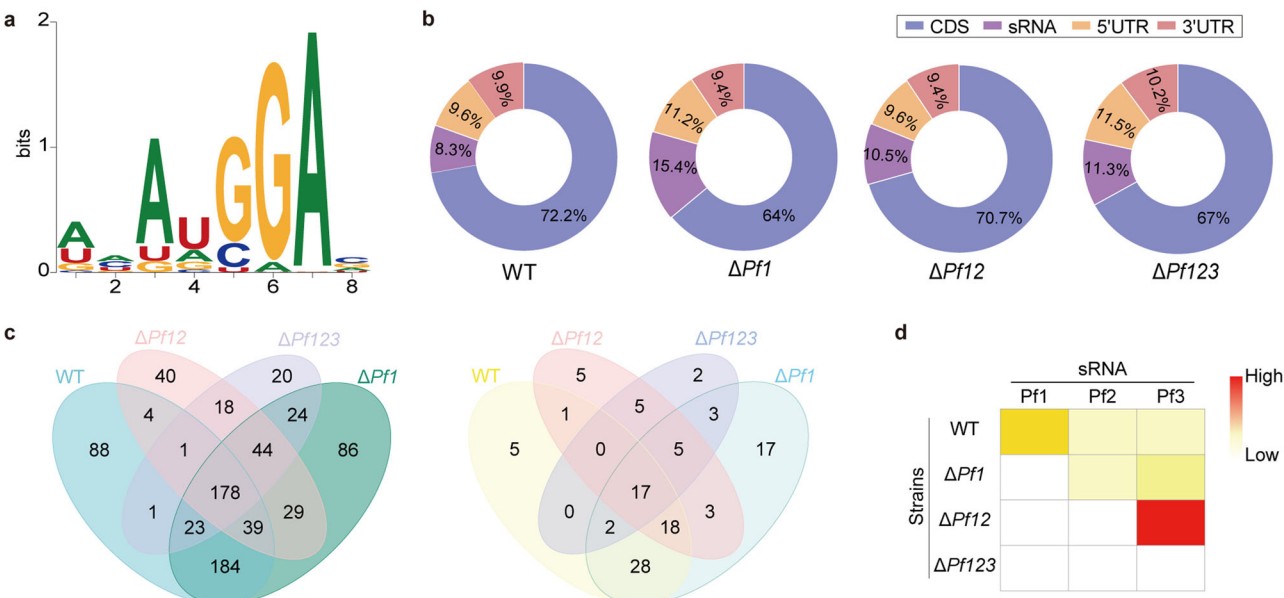

**Fig. 3 | RIP-seq analysis across the WT and Pf sRNA mutants. a** Visualization of the CsrA binding motif of all peaks **b** Pie charts of categories of CsrA targetome in each strain. **c** Venn diagrams of the distribution of peaks annotated as genes (left) and sRNAs (right) across four strains. Peaks without annotations of gene or sRNA are not displayed. The full list of targets for the Venn diagram is available in Supplementary Data 2. **d** A heatmap representing the relative abundance of Pf sRNAs (Pf1, Pf2, Pf3) in the CsrA targetomes across strains.

on late-exponential phase lysates of *csrA*-3×FLAG strains and, as control, their same lysates that have not been incubated with anti-FLAG beads. The conversion of co-purified RNA into cDNA and subsequent deep sequencing yielded an effective mapping of 98.99% of the sequencing reads onto the genome, ranging from 13.68 to 22.33 million reads per library. Over 700 RIP-seq peaks showed significant enrichment ($q < 0.01$) in the 3×FLAG-CsrA samples compared to the input control (Supplementary Data 2). These peaks corresponded to 779 protein-encoding genes and 111 putative sRNAs, representing 18.9% and 15.4% of the total genes and sRNAs predicted in the genome of *P. fuliginea* BSW20308, respectively.

The detailed sequence and structural motif analysis highlighted the prevalence of an "AuGGA" motif (Fig. 3a). The enrichment of this "AuGGA" motif within the RIP-seq peaks, in comparison to the overall genome sequence, was statistically significant ($P < 0.0001$). The CsrA RIP-seq peaks associated genomic features were summarized in the Fig. 3b. The majority (84.6–91.7%) of CsrA binding sites were concentrated in mRNA regions, with a substantial fraction (64–72.2%) specifically localized to coding sequences. The remaining peaks were distributed across the 5′ untranslated regions (UTR) and 3′ UTR of mRNAs, consistent with the fact that CsrA can bind to both regions to modulate mRNA translation and/or RNA stability[30].

A substantial number of different potential targets ranging from 77.2% and 84.7% of the total mRNAs and sRNAs peaks, respectively, were identified among the WT, Δ*Pf1*, Δ*Pf12*, and Δ*Pf123* strains (Fig. 3c). The results supported our hypothesis that the presence of different number of Pf sRNAs had an influence on the CsrA targetomes. Intriguingly, the RIP-seq peaks contained a considerable amount of sRNA (8.3–15.4%), which is much higher than that observed in *E. coli* and other strains (around 2–5%)[1,7,9]. Furthermore, Pf sRNAs were detected in the RIP-seq results as the top significantly enriched peaks, indicating a prevalent binding of these Pf sRNAs to CsrA in vivo. Interestingly, a compensatory effect was observed among the three Pf sRNAs. Specifically, Pf1 showed the highest enrichment in the WT strain, followed by Pf2 and Pf3 as the second and third highest peaks, respectively (Fig. 3d). In the Δ*Pf1* mutant strain, Pf2 and Pf3 emerged as the top two enriched peaks (Fig. 3d), whereas in the Δ*Pf12* mutant strain, Pf3 became the most enriched peak.

The enrichment of genes in KEGG pathways corresponding to all CsrA RIP-seq peaks was systematically analyzed (Supplementary Fig. 5). The knockout of sRNAs affected the CsrA targetomes in two ways: by changing the enriched categories of KEGG pathways and by altering the level of involvement of specific genes within the same pathways. Specifically, fewer pathways were enriched in the mutants compared to the wild strain (17–25

https://doi.org/10.1038/s42003-025-07780-y                                                                **Article**

in the mutants versus 28 in the wild strain). Notably, some pathways such as DNA replication and repair and propanoate metabolism, were exclusively enriched in the wild strain. Furthermore, even for pathways that were enriched across all strains, there were variations in the number and types of genes contributing to specific pathways. Detailed information on the enriched functions of genes with CsrA RIP-seq peaks and complete pathway results were shown in the Supplementary Data 3. Several of these enriched functional pathways corresponding to the known functions of CsrA and potential new targets were further analyzed in subsequent sections.

### Pf sRNAs modulated CsrA interaction with cell motility-related genes in *P. fuliginea* BSW20308

*P. fuliginea* BSW20308 contained a flagellar synthesis gene cluster comprising more than 50 genes, involved in hierarchy regulation and biosynthesis, similar to those found in *P. aeruginosa* and *V. cholerae* (Fig. 4a and Supplementary Fig. 6)[25]. Notably, RIP-seq analysis revealed the presence of 30 out of 50 flagellar synthetic genes, with 18 genes belonging to classes II-IV consistently detected across all four strains. These 18 shared genes encode vital components essential for flagellar synthesis, including the export apparatus, rod, hook, and critical regulatory factors (FlrB and FlrC) (Fig. 4a, b). Consequently, CsrA protein exerted regulation on both structural genes and regulation genes of flagellar biosynthesis, suggesting direct and indirect regulatory function on flagellar synthesis in *P. fuliginea* BSW20308.

The remaining 12 flagellar biosynthetic genes were observed in some of the four strains, including the master regulator *fleQ* and Class III regulator *fliA*, as well as 10 structural biosynthetic genes. Notably, 10 (including *fliA*, *fliD*, *flaG*, *fliN*, *fliM*, *fliC*, *flgL*, *flgM*, *flgT*, and *flhG*) out of the 12 genes were only detected in the Pf sRNA mutant strains while absent in the WT strain, suggesting that the deletion of Pf sRNAs influenced the CsrA targetomes. For instance, *fliA* was exclusively found in the Δ*Pf123* strain, implying that the deletion of all Pf sRNAs enabled CsrA protein to bind to *fliA* mRNA, thereby affecting class IV flagellar synthesis (Supplementary Data 4).

Remarkably, the number of flagellar biosynthetic genes detected in the WT, Δ*Pf1*, Δ*Pf12*, and Δ*Pf123* strains increased gradually, reaching 19, 25, 26, and 28 genes, respectively. This trend indicated that as the number of Pf sRNAs knocked out increased, CsrA appeared to gain greater proficiency in regulating the genes involved in flagellar synthesis. These target genes were known to be positively regulated by CsrA in *E. coli*[30–33]. Therefore, we hypothesized that the greater number of target genes identified would correlate with an enhanced motility in the *Pseudoalteromonas* strains. Consistent with our hypothesis, we observed a gradual increase in swimming ability among the four strains, following the order of the Δ*Pf123*, Δ*Pf12*, Δ*Pf1*, and WT strains, as demonstrated by the increasing spreading area on 0.3% agar plates (Fig. 4c and Supplementary Fig. 7). Additionally, we confirmed through EMSA that *fleQ* mRNA could specifically bind to CsrA in vitro (Fig. 4d).

### Pf sRNAs influenced CsrA-mRNA interactomes for cell envelope proteins

The RIP-seq analysis revealed a multitude of mRNAs encoding cell envelope proteins related to the Tol/Pal system, sigma E (RpoE) envelope stress pathway, and phage shock protein (PSP) system, among others (Supplementary Data 4). Among the five core proteins of the Tol/Pal system (TolQ, TolR, TolA, TolB and Pal), the RIP-seq analysis pinpointed *tolA*, *tolB*, and *pal* mRNAs as CsrA targets, with *pal* mRNA capable of binding to CsrA protein in vitro and competing with the three sRNAs (Pf1, Pf2, and Pf3) for CsrA binding. (Fig. 5 and Supplementary Fig. 8). Peaks of *tolB* and *pal* were detected in all the four strains, while the peak of *tolA* was only observed in the WT and Δ*Pf1* strains (Fig. 5). Several key genes related to sigma E (RpoE) envelope stress pathway were detected, with *rseB*, *clpX*, and *clpP* being present in the CsrA targetomes across four strains, whereas *rseA* and *rpoE* were exclusively observed in the Δ*Pf123* strain (Fig. 5). The *PspABC* operon of the PSP system, engaged in the reception of external signal stimulation, was detected in the RIP-seq data. Specifically, the CsrA binding peak was

concentrated upstream of *pspABC* mRNA in the WT, Δ*Pf1*, Δ*Pf12* strains but absent in Δ*Pf123* strain (Fig. 5 and Supplementary Data 4). However, PspF, a sigma 54-dependent activator of *pspABC* transcription, was absent from our RIP-seq data (Fig. 5). Therefore, CsrA probably influenced the signal transduction of BSW20308 by directly regulating the *pspABC* mRNA.

Additionally, mRNAs encoding other proteins involved in envelope stress response, transport, and other functions were differentially detected in the RIP-seq data across strains. These included the β-barrel assembly machinery (BAM) complex, outer membrane proteins (OMPs), the Ton, Mur, and Fts complexes (Fig. 5 and Supplementary Data 4). These findings highlighted distinct regulation patterns for key genes like *bamE*, *ompH*, *murE/F/G*, *ftsI*, and *ftsW*, which were present in one or a subset of the four strains under investigation.

### The type VI secretion system and its differential regulation across strains

The Type VI Secretion System (T6SS), a prevalent transmembrane complex to deliver effectors to adjacent cells, plays a crucial role in competition, environmental adaptation, and virulence in many pathogens[34,35]. There are more than 18 genes associated with the T6SS in *P. fuliginea* BSW20308 (Fig. 6a and Supplementary Data 4), encoding 13 core conserved components (TssA-M) for the apparatus assembly and several accessory proteins such as TagH, TagF, and PAAR[36]. Six of them were detected in the RIP-seq data, encoding tube proteins (VgrG and Hcp), a substrate protein (TssK), a membrane complex component (TssJ), an ATPase (TssH), and the accessory protein TagH involved in initial activation of the T6SS assembly[37,38] (Fig. 6b). Specifically, *tagH*, *tssJ*, and *tssK* genes were consistently detected across all four strains, while *hcp* (*tssD*), *tssH*, and *vgrG* (*tssI*) were exclusively found in the WT and Δ*Pf1* mutant strains (Fig. 6b and Supplementary Data 4).

Upon analyzing the conserved domains associated with the T6SS, five potential T6SS effectors were predicted in the *P. fuliginea* BSW20308 genome (Supplementary Fig. 9). Among them, three (WP_007377080, WP_058230555, and WP_007377877) were predicted to contain RhsA domains, alongside other conserved domains. Specifically, WP_058230555 possessed a ClyA-like domain, which is associated with pore-forming activity, while WP_007377080 contained a PoNe_PAAR-like domain, indicative of a polymorphic nuclease function (Supplementary Fig. 9). Intriguingly, WP_007377877 lacked discernible functional domains except the RhsA domain. Furthermore, the hypothetical protein WP_033023891, possessing a conserved Tse2 ADP-Ribosyltransferase Toxin (ADPRT) domain, was predicted to be a Tse2-like effector. This ADPRT domain, previously identified in a T6SS effector of *P. aeruginosa*[39], is known to transfer phosphate ribose from ribonucleotides to receptor proteins, thereby interfering with the function of receptor proteins and leading to cell death. Lastly, the putative effector WP_007377695 containing a Vgr_GE domain could function as a muramidase[40].

Based on the RIP-seq data, three of the putative effectors, WP_033023891, WP_007377080, and WP_007377877, were detected as potential CsrA targets (Supplementary Data 4). Specifically, WP_033023891 was commonly detected across all four strains analyzed, while the other two were exclusively detected in the WT and Δ*Pf1* strains. Further analysis revealed the presence of putative CsrA binding sites, characterized by GGA motifs and predicted to form stem-loop structures, upstream of the start codons of WP_033023891 and WP_007377080 (Fig. 6c). The results of EMSA confirmed that CsrA bound specifically to the 5'UTRs of WP_033023891 and WP_007377080 mRNAs in vitro (Fig. 6d and Supplementary Fig. 8). As for WP_007377877, a potential CsrA binding site was identified within its coding region, and EMSA results demonstrated that the mRNA corresponding to this region could bind to CsrA in vitro (Fig. 6c, d).

### Pf sRNAs modulated the interaction between CsrA and mRNA of regulators associated with biofilm formation

Biofilm formation is a multifaceted process that encompasses the production of extracellular polysaccharides (EPS), curli, and many other

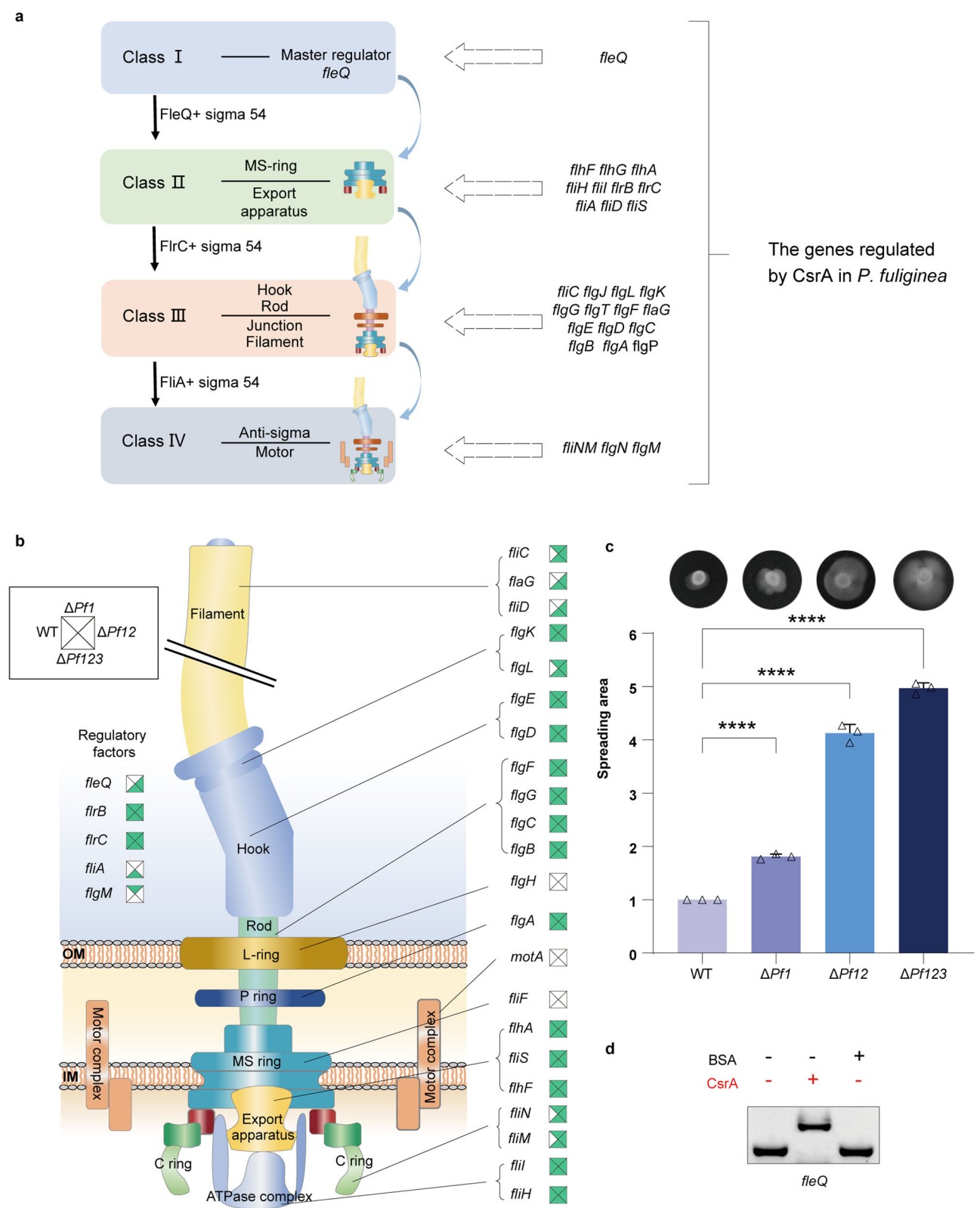

components. This intricate process is mainly controlled by the Quorum Sensing (QS) system, the secondary messenger c-di-GMP (bis-(3′-5′)-cyclic dimeric guanosine monophosphate) and sigma factors like RpoE and RpoS[41]. However, the crosstalk between the Csr system and other regulatory systems related to biofilm formation is under-investigated. To understand the impact of Pf sRNAs on static biofilm formation of *P. fuliginea*

BSW20308, biofilm was quantitatively measured and compared for the four strains. A significant reduction in biofilm formation was observed in the Δ*Pf12* and Δ*Pf123* strains, whereas the Δ*Pf1* strain exhibited no notable decrease compared to the WT strain (Fig. 7a).

However, the genes directly responsible for EPS (*pelA-D*) and curli (*csgD-G*) biosynthesis present in *P. fuliginea* BSW20308 genome were

**Fig. 4 | Pf sRNAs modulate CsrA interaction with cell motility-related genes.**
**a** The putative model of flagellar hierarchy assembly and the flagellar genes detected in the CsrA targetomes. The flagellar hierarchy assembly proceeds as follows: 1) The master regulator, FleQ (Class I), activates the transcription of Class II genes with the assistance of RpoN ($\sigma^{54}$). 2) Class II genes encode components essential for export apparatus, the MS-ring, two component regulators FlrBC and FliA. FlrBC activate the expression of class III genes, while FliA controls the transcription of Class IV genes. 3) Phosphorylated FlrC activates Class III genes responsible for synthesizing the hook (*flgDE*), L-ring (*flgH*), P-ring (*flgA*), rod (*flgBCFG*), kook filler junction (*flgKL*), and filament (*fliC*). 4) Class IV genes transcribe motor-related proteins and

anti-sigma (FlgM) in the presence of FliA. Our RIP-seq data revealed 30 flagellar synthetic genes, with some displaying specific binding to CsrA in specific strains (full list provided in Supplementary Data 4). **b** The detection of flagellar related genes in the CsrA targetomes. Squares near gene names show the detection in the WT (left), Δ*Pf1* (top), Δ*Pf12* (right), and Δ*Pf123* (bottom) strains, with green representing a positive result and white representing a negative result. **c** Swimming ability of the WT, Δ*Pf1*, Δ*Pf12*, and Δ*Pf123* strains indicated by spreading area on 0.3% agar plates. Plotted is the mean ± s.e.m (****$P < 0.0001$ using Student's *t*-test). **d** EMSA of 1 μM CsrA binding to mRNAs of *fleQ* with 1 μM BSA as the control.

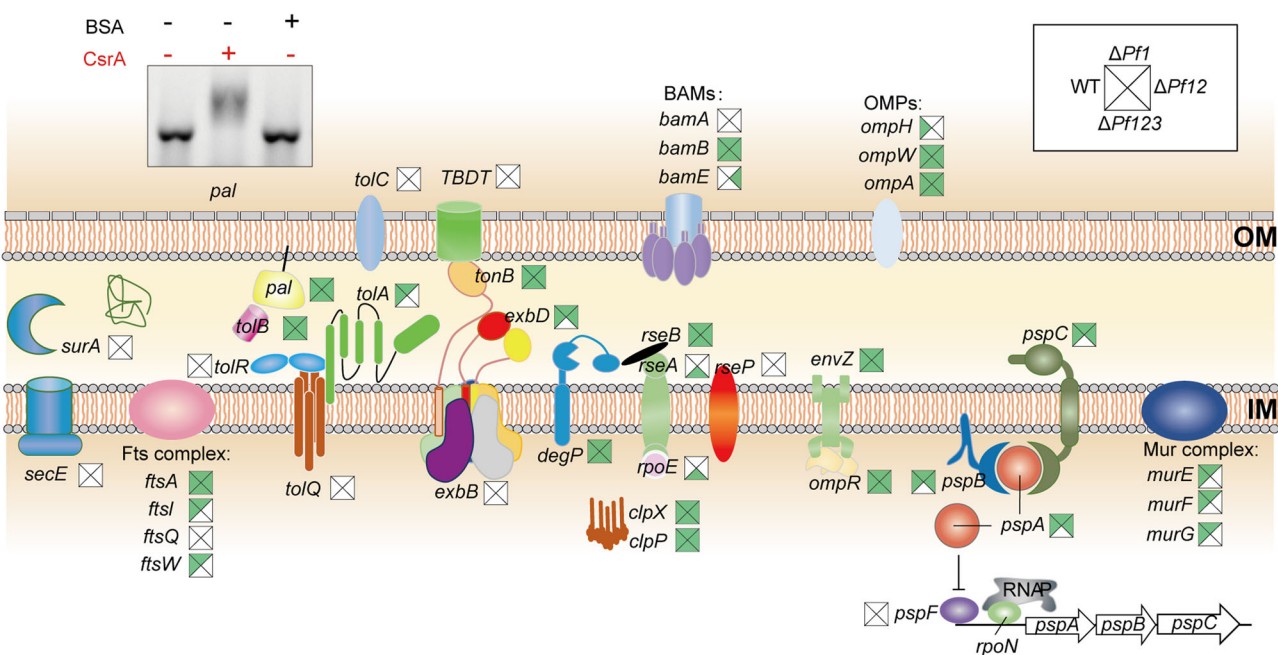

**Fig. 5 | Influence of Pf sRNAs on cell envelope proteins across strains.** Squares near gene names show the detection in the WT (left), Δ*Pf1* (top), Δ*Pf12* (right), and Δ*Pf123* (bottom) strains, with green representing a positive result and white

representing a negative result. The inset in the upper left shows an electrophoretic mobility shift assay (EMSA), demonstrating the in vitro binding of CsrA protein to *pal* mRNA. IM inner membrane, OM outer membrane.

absent from the CsrA targetomes, suggesting that the Csr system probably regulates the regulators involved in biofilm formation at a higher hierarchical level. Indeed, genes encoding sigma factors RpoE and RpoS, and the diguanylate cyclase that synthesizes c-di-GMP, were detected in the CsrA targetomes (Fig. 7b and Supplementary Data 4). Additionally, other genes encoding transcription regulators (Fis and H-NS) and two-component system (EnvZ/OmpR) related to biofilm formation were also found in the CsrA targetomes, with both *ompR* mRNA and *h-ns* mRNA capable of binding to CsrA protein in vitro (Fig. 7c and Supplementary Fig. 8). During biofilm formation, regulators played distinct roles. Among them, c-di-GMP, EnvZ/OmpR, and RpoE were known to enhance biofilm formation, whereas RpoS, Fis, and H-NS had negative effects (Fig. 7b)[42–44]. Notably, all CsrA targets associated with biofilm formation were commonly detected across strains, except for rpoE which was exclusively found in the Δ*Pf123* strain (Fig. 7b). Therefore, biofilm formation involves a complex regulatory network, which complicates the explanation for decreased biofilm formation observed in Pf sRNA mutants.

### Potential specific targets of Csr system in *P. fuliginea* BSW20308
To identify specific targets of the Csr system in *P. fuliginea* BSW20308, we analyzed 616 gene targets detected through RIP-Seq (Supplementary Data 5), of which 138 were classified as specific targets compared to *E. coli*[1,24].

Among these, several represent previously unreported targets, including D172_RS01480 (nemA), involved in the degradation of toxic compounds[45,46]; *D172_RS00585* (cytochrome c), potentially involved in electron transport within the respiratory chain and *D172_RS04550* (yfiA), implicated in the regulation of translation efficiency[47]. Additionally, several hypothetical proteins were identified, representing unexplored functional pathways (Supplementary Data 5). Some of these hypothetical proteins could be linked to previously established roles of the Csr system, such as the regulation of T6SS-related genes. For instance, a previously studied protein, WP_033023891, which is a potential T6SS effector, was shown to directly bind to CsrA in vitro (Fig. 6d). These findings highlight the unique targets of the Csr system in *P. fuliginea* BSW20308 compared to *E. coli*, underscoring species-specific regulatory mechanisms and adaptations to its environmental niche.

### Pf sRNAs modulated the interaction between CsrA and other sRNAs
There were hundreds of sRNAs in *P. fuliginea* BSW20308 predicted in our previous RNA-seq studies[24,25], 111 were detected in this study as the putative CsrA interactors including the three Pf sRNAs (Supplementary Data 6). Various number of putative sRNAs were detected in CsrA targetomes, including 71, 93, 54, and 34 found in the WT, Δ*Pf1*, Δ*Pf12*, and Δ*Pf123*

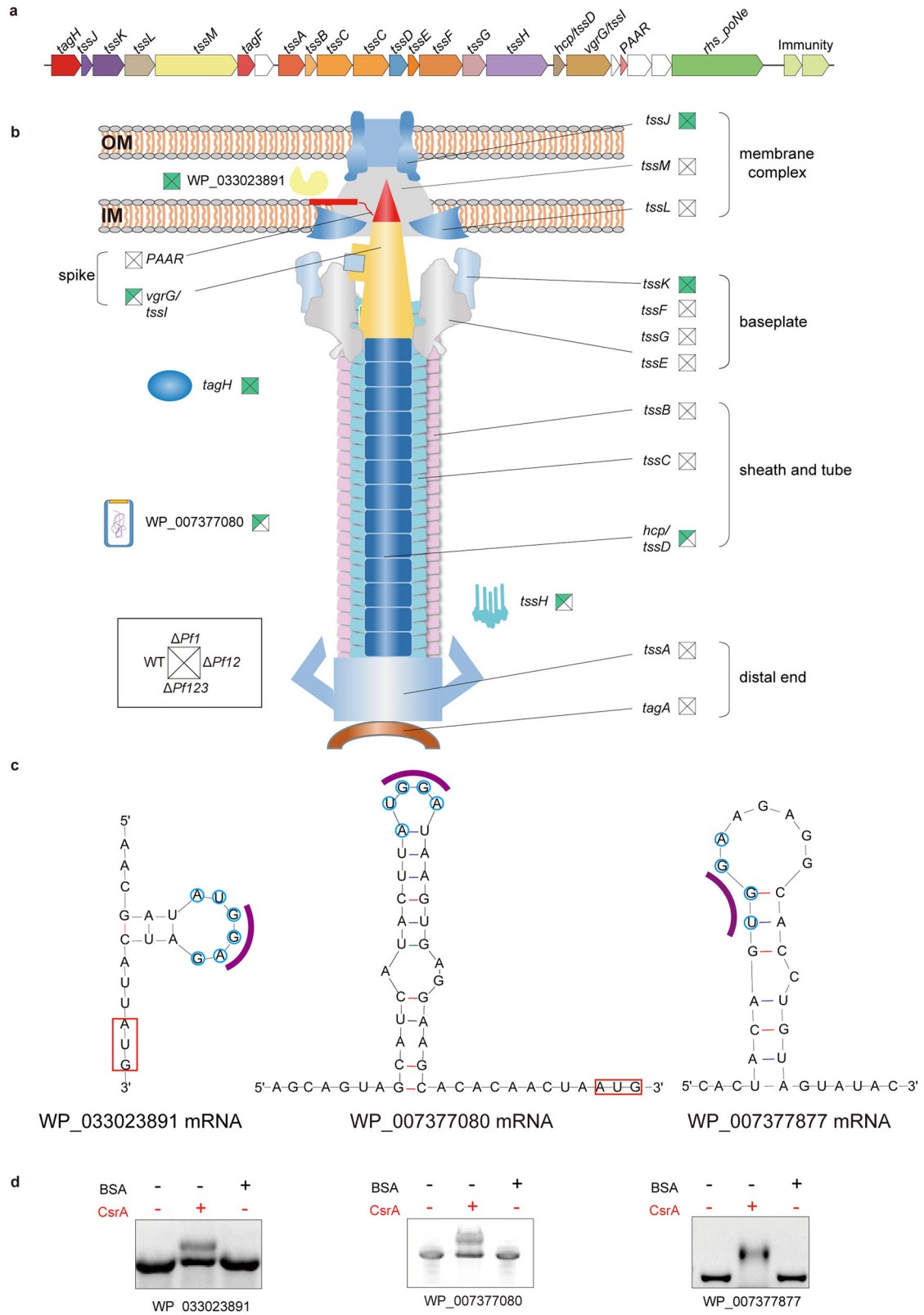

**Fig. 6 | The type VI secretion system and its differential regulation across strains.**
**a** Schematic representation of the T6SS gene cluster in *P. fuliginea* BSW20308. Genes are depicted as arrows pointing in the direction of transcription. The gene names are marked above. White indicates unknown genes. The term "immunity" refers to genes encoding for putative immunity proteins that neutralize T6SS effectors. **b** The structural organization of T6SS and the related genes detected in the CsrA targe-tomes across strains. Squares near gene names show the detection in the WT (left),

ΔPf1 (top), ΔPf12 (right), and ΔPf123 (bottom) strains, with green representing a positive result and white representing a negative result. **c** Predicted CsrA-mRNA binding sites for mRNAs of WP_033023891, WP_007377080, and WP_007377877. Putative CsrA binding sites are marked with blue circles, while GGA motifs involved in binding are indicated with purple arcs. Red squares mark the location of start codons. **d** EMSA of 1 μM CsrA binding to mRNAs of WP_033023891, WP_007377080, and WP_007377877, with 1 μM BSA as the control.

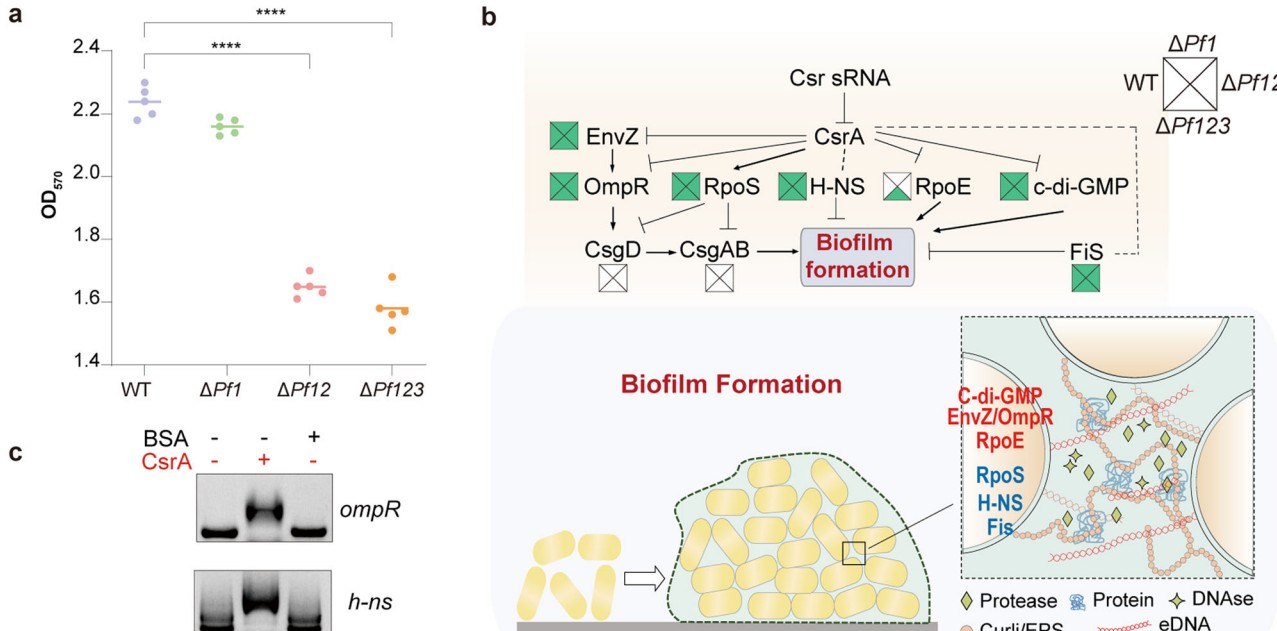

**Fig. 7 | Influence of Pf sRNAs on biofilm formation across strains. a** Biofilm formation capabilities across four strains. Quantification of biofilm formation was determined using the crystal violet staining assay (n = 5). Statistically significant differences are indicated by asterisks (****), with $P < 0.0001$ denoting highly significant differences in biofilm formation between the WT strain and the *Pf* sRNA mutants. **b** Regulation of genes involved in multiple regulators associated with the biofilm formation by the Csr system in *P. fuliginea* BSW20308. Squares near gene names show the detection in the WT (left), Δ*Pf1* (top), Δ*Pf12* (right), and Δ*Pf123* (bottom) strains. The dashed line indicates the as-yet-unidentified regulatory effect. Regulators in red are known to play a positive role in biofilm formation, while those in blue denote a negative role. **c** EMSA of 1 μM CsrA binding to mRNAs of *ompR* (top) and *h-ns* (bottom), with 1 μM BSA as the control.

strains, respectively (Fig. 3c and Supplementary Data 6). A decreasing trend of sRNAs was detected along with the deletion of one to three Pf sRNAs. Among them, only 17 sRNAs were commonly shared by the four strains, accounting for 18–50% sRNAs detected in each strain. Using EMSA, we verified that the core sRNA, sRNA260, could bind to the CsrA protein in vitro (Supplementary Fig. 10).

Since these putative sRNAs were detected in the CsrA targetomes, the presence of the GGA motif was analyzed. Zero to four GGA motifs were detected for these putative sRNAs, except the three Pf sRNAs which contained 25, 21, and 9 GGA motifs (Supplementary Data 6). Out of the 111 detected sRNAs in the CsrA targetomes, 35 sRNAs contained zero GGA motifs. The majority of the remaining sRNAs contained one GGA motif.

Based on the target prediction in silico, 69 of the 111 sRNAs were predicted to be cis-encoded sRNAs, forming perfect base-pairing with various targets belonging to T6SS, flagellar biosynthesis, stress responses, transposase family, and alcohol metabolism, etc. (Supplementary Data 6). Notably, the putative antisense sRNA0654 shared by the four strains was predicted to regulate CsrA mRNA. Nevertheless, putative imperfect base-pairing targets were also predicted by RNAphybrid and RNAplex for those potential cis-encoded sRNAs (Supplementary Data 6). The remaining 41 sRNAs detected in the CsrA targetomes were predicted to form imperfect base-pairing with multiple potential targets, a characteristic of trans-encoded sRNAs (Supplementary Data 6). In particular, sRNA0652, shared by all the four strains, was predicted to target 236 potential genes. Unexpectedly, the three Pf sRNAs were also predicted to form imperfect base pairing with a few mRNA targets. It seems that the Csr system probably formed complex crosstalk with the Hfq system.

The potential binding of RBPs to the commonly detected sRNA candidates, sRNA0652 and sRNA0654, was predicted using AlphaFold 3. Both sRNA0652 and sRNA0654 were shown to bind to Hfq hexamer and CsrA dimers at different sequence regions (Figs. 8 and 9). RNA sequences bond to the positively charged surfaces (in blue) of CsrA dimmers formed by the first β-sheet of one monomer and the last β-sheet of the other monomer, according to the predicted 3D structure and interactions. This prediction is consistent with previous structural studies in *E. coli* and *P. fluorescens*[48,49]. However, since there was zero GGA motif in both sRNAs, the sequences binding to CsrA were different, including CGA, UAU, CCU, and UCA (Fig. 9). Both U-rich sequences of sRNA0652 and sRNA0654 were shown to interact with the recessed channel on the Hfq proximal face (Fig. 9), consistent with the previous report of sRNA Rydc-Hfq structure[50].

## Discussion

The overall discoveries have been summarized in Fig. 10 and discussed below. This study successfully identified two additional Csr sRNAs, Pf2 and Pf3, in *P. fuliginea* BSW20308. Consequently, a total of three Csr sRNAs have been discovered so far in this strain, representing the first comprehensive survey of the Csr system in *Pseudoalteromonas* and bacteria inhabiting polar regions. Homologs of these three Pf sRNAs coexist in over one-third of *Pseudoalteromonas* genomes, demonstrating that the presence of multiple Csr sRNAs is highly prevalent in this genus. Following the discovery of two to four copies of Csr sRNAs in various species, including *E. coli*[12], *P. aeruginosa*[15], *P. fluorescens*[16–18], *V. cholerae*[13] and *V. tasmaniensis*[14], *Pseudoalteromonas* becomes the latest genus known to possess multiple Csr sRNAs. It seems that multiple Csr sRNAs are probably the most common phenomenon in the bacterial Csr systems, further supporting the idea that the Csr system is very robust due to the presence of multiple Csr sRNAs with high affinity.

In general, the evolution of sRNAs is dynamic due to the varied selective pressures and relatively unrestrictive structural requirements, which contributes to high species variation and poses challenges in tracing sRNAs[41]. Notably, Pf1 and Pf2 seem to be confined to *Pseudoalteromonas*, as no homologs were detected in other genera. This suggests the de novo emergence of these two Pf sRNAs in the *Pseudoalteromonas* genus. Unexpectedly, Pf3 exhibits homologs in animals such as zebrafish. Interestingly, the aligned sequences of the zebrafish genome contained TAA-rich or AAT-rich repeats, which might be a special class of microsatellites[51] or some form of retrotransposition[52]. While it remains unclear whether these eukaryotic homologs are evolutionary or functionally related to Pf3, this finding poses

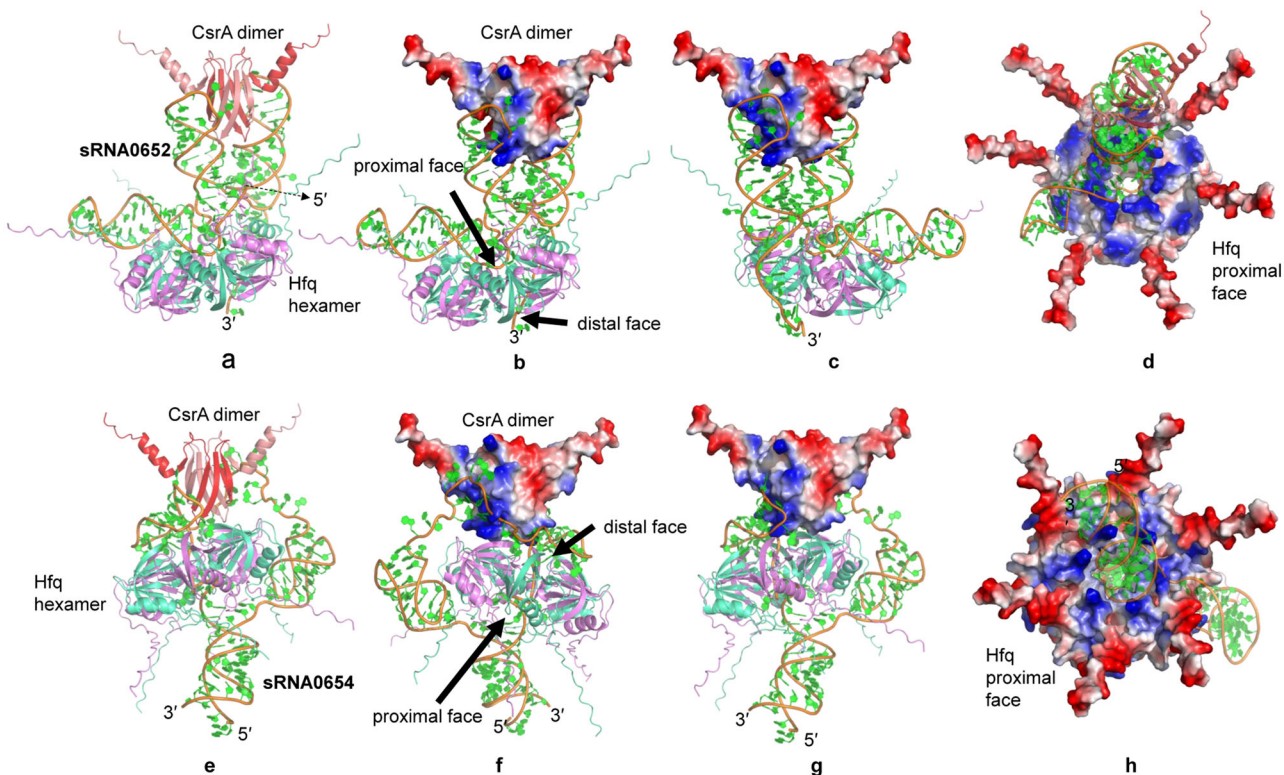

**Fig. 8 | Predicted structures and interactions of CsrA, sRNA and Hfq.** Ribosomal binding proteins CsrA and Hfq interacting with sRNA0652 (**a–d**) and sRNA0654 (**e–h**).

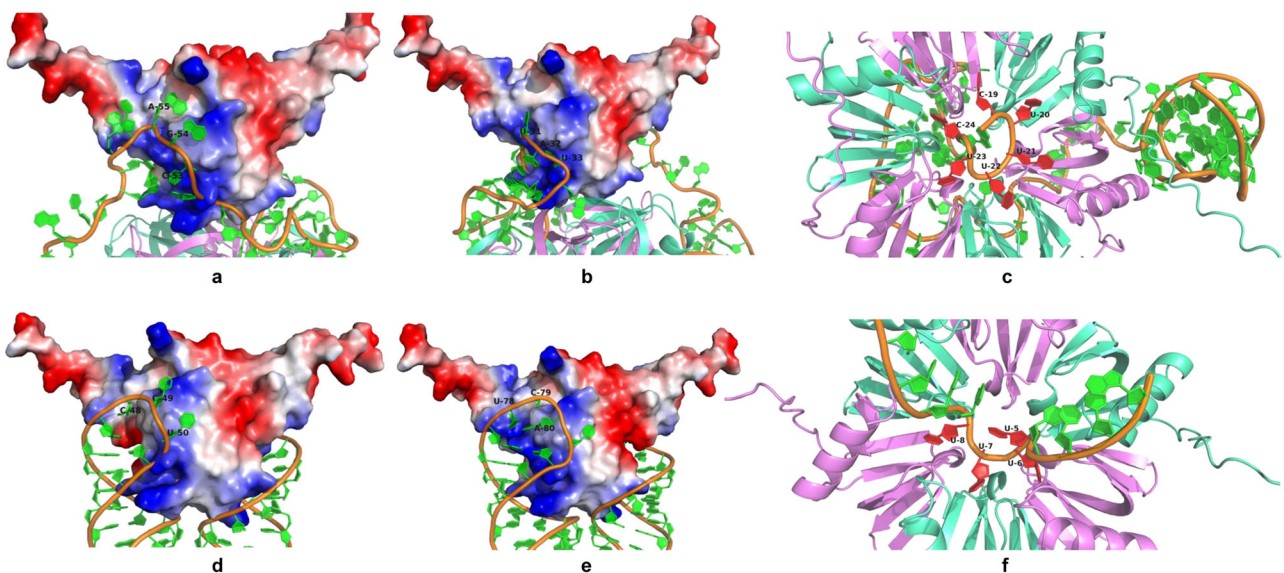

**Fig. 9 | Predicted binding sites of sRNAs to CsrA and Hfq.** Binding sequences of sRNA0652 to CsrA (**a**, **b**) and Hfq (**c**); sRNA0654 to CsrA (**d**, **e**) and Hfq (**f**).

intriguing questions regarding the evolution of Pf3 in *Pseudoalteromonas*. Further research on this topic is necessary to elucidate the evolutionary history of Pf sRNAs and to determine if there is any relationship between bacterial sRNAs and eukaryotic sequences.

The presence of multiple Csr sRNAs has been considered to be functionally redundant according to previous studies[13,14,18]. However, despite the non-lethality of the loss of one to three Pf sRNAs, the growth and physiological responses are influenced to varying degrees. Notably, the greater number of Pf sRNAs lost correlates with a more severe impact. This impact is contingent upon environmental conditions, as growth inhibition is more pronounced in high sucrose conditions compared to high salinity for Pf

sRNA deletions. This suggests that, rather than functional redundancy, Pf sRNAs exhibit functional complementarity and are crucial for maintaining normal function and growth. This highlights the intricate interplay between Pf sRNAs and underscores their essential contribution to the overall fitness and stress tolerance of *Pseudoalteromonas* strains. Remarkably, the expression of Pf2 and Pf3 increased (though not significantly) in previous RNA-seq analyses of ∆*Pf1*[22], accompanied by a significant rise in the relative abundance of Pf2 and Pf3 with the CsrA protein (Fig. 3d). This further implies a compensatory strategy among the three Pf sRNAs, enhancing the robustness of the system. Moreover, Csr sRNAs could be differentially regulated in response to various signals as shown in other species,

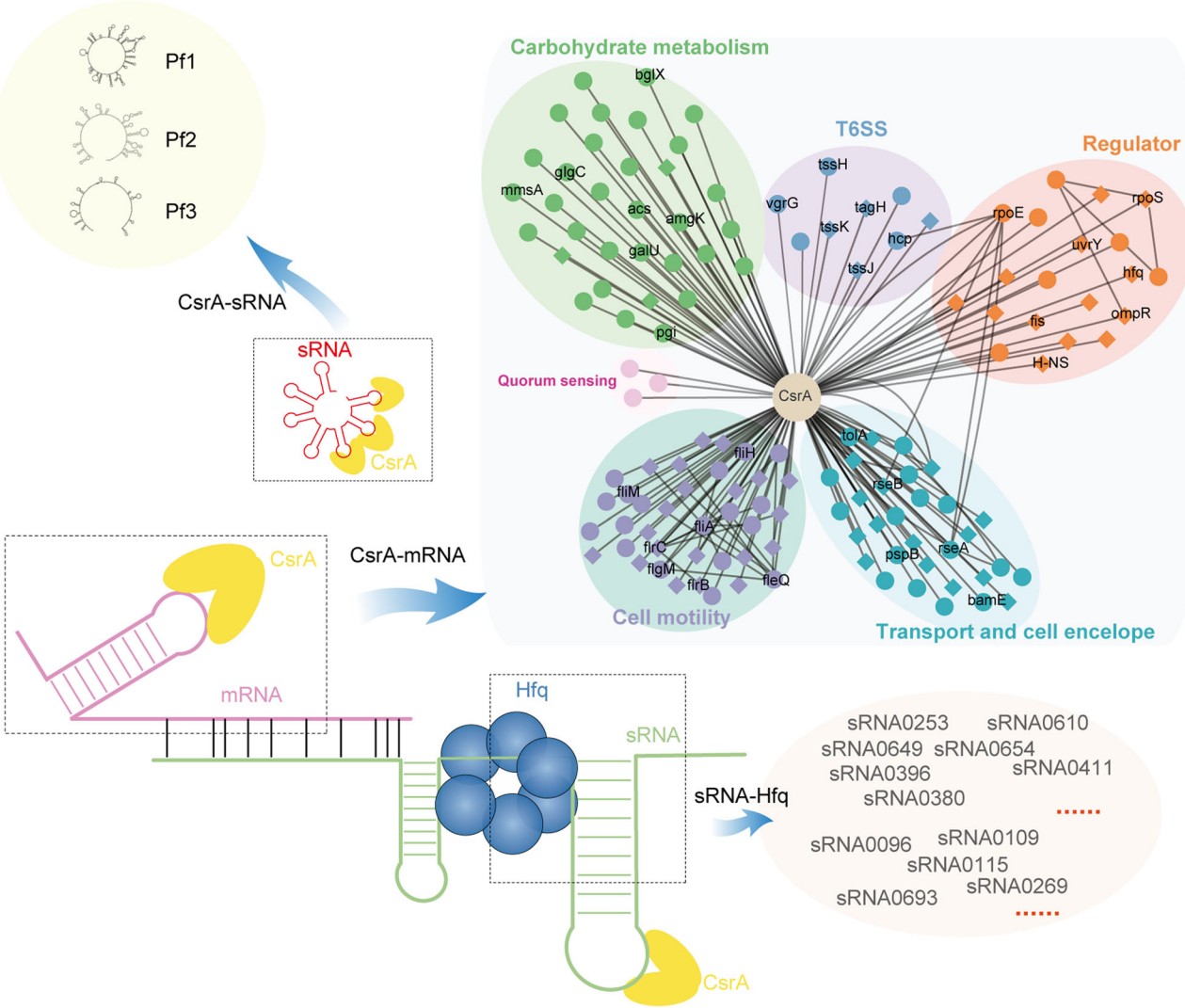

**Fig. 10 | Model of crosstalk among CsrA, sRNA, mRNA, and Hfq.** The CsrA protein functions as a global regulator, binding to mRNA related to the type VI secretion system, cell motility, and cell envelope, etc. Circles denote genes that exhibit consistent binding to CsrA protein in WT and Δ*Pf* mutants. Diamonds indicate genes with variable CsrA binding across the strains. (top right). Meanwhile, CsrA can bind to sRNAs, including red labeled "sponge sRNAs" (e.g., Pf1, Pf2, and Pf3 at top left) and green labeled "base-pairing sRNAs" (bottom). Most trans-encoded sRNAs necessitate the assistance of Hfq protein for pairing with their targets.

demonstrating the necessity of multiple Csr sRNAs and the flexibility of the Csr systems. For instance, in *V. tasmaniensis*, four *csrB* genes are distinctively regulated[14], while in *Salmonella enterica*, CRP-cAMP differentially regulates CsrB and CsrC, with little effect on the former and transcriptional inhibition of the latter[53]. It is plausible that a similar mechanism exists in *P. fuliginea* BSW20308, which merits further investigation.

Consistent with the observed variations in physiology and growth, the loss of varying numbers of Pf sRNAs exerts a significant impact on the CsrA targetomes. Our comparative RIP-seq analysis uncovered numerous distinct targets in the three mutant strains (Δ*Pf1*, Δ*Pf12*, and Δ*Pf123*) compared to the WT strain. Notably, the knockout of Pf sRNAs not only altered the number but also the composition of the CsrA targetomes, indicating a complex interplay rather than a mere quantitative change. Complete absence of Pf sRNAs did not exhibit broader binding to other available RNA targets due to the loss of its primary antagonists. Such a shift could potentially result in changes to the expression levels of other CsrA-regulated genes, including those of previously uncharacterized sRNAs or certain mRNAs that might act as competitive decoys to sequester CsrA. As shown in the results, genes involved in crucial processes such as motility, signal transduction, and T6SS were present differentially in the CsrA targetomes

across the four strains. In the case of flagellar biosynthetic genes, an increasing number of genes were detected in the CsrA targetomes of the WT, Δ*Pf1*, Δ*Pf12*, and Δ*Pf123* strains, correlated to their increasing motility. It seems that the knockout of Pf sRNAs could increase the availability of CsrA to flagellar-related targets in this case. However, the total number of mRNA targets basically follows an opposite trend, except for the highest number detected in Δ*Pf1*. Despite different biofilm production was measured across the four strains, key regulators related to biofilm formation were detected in all CsrA targetomes. For T6SS, putative effectors were differentially detected in the CsrA targetomes. These findings further highlight the complex effects of various Pf sRNAs on the CsrA targetomes, thereby influencing the survival and adaptation of *P. fuliginea* BSW20308, consistent with our previous study on the knockout of Pf1 sRNA[24]. This study underscores the intricacy of the Csr system and the importance of the presence of multiple Csr sRNAs.

Another intriguing finding of this study is the detection of numerous putative sRNA targets of CsrA. The interplay between these sRNAs and CsrA, and consequently, the outcomes of this interaction, are likely intricate (Fig. 10). One plausible scenario is that these sRNAs might act as alternative antagonistic or "sponge" sRNAs, in addition to traditional Csr sRNAs. For

instance, some sRNAs such as Spot42, GadY, McaS, MicL, and SgrS have been documented to bind CsrA directly, thereby sequestering and antagonizing its activity[9,54,55]. However, their binding affinity and efficiency are much lower than that of the classical Csr sRNAs, primarily due to the presence of only one or two GGA motifs in these sRNAs, or even the absence of typical GGA motif such as MicL[56]. The majority of putative sRNAs detected in the CsrA targetomes of *P. fuliginea* BSW20308 contain zero to one GGA motif. Therefore, the significance of these sRNAs in antagonizing CsrA remains to be further explored. Another potential consequence of the binding of these sRNAs to CsrA is enhanced stability. As exemplified by Spot42, a Hfq-binding antisense sRNA, it can also bind to CsrA, thereby preventing its degradation by blocking the RNase E binding site[57]. Further verification is required to ascertain whether there are any other sRNAs analogous to Spot42 within the CsrA targetomes of *P. fuliginea* BSW20308.

Although experimental verification should be performed to valid the presence of the putative sRNAs and their potential targets, the detection of such a significant number of trans-encoded base-pairing sRNAs in the CsrA targetomes indicates the existence of complex crosstalk between the Csr system and the Hfq system (Fig. 10). This is also hinted by the 3D structures showing possible interactions between sRNAs, Hfq and CsrA (Figs. 8 and 9). These two RNA-binding systems have been studied mostly independently due to their distinct binding and function mechanisms, but recent studies have uncovered that they crosstalk[20,58]. For example, in *E. coli*, certain sRNAs like DsrA and FnrS, which require the assistance of the RNA-binding protein Hfq to facilitate their binding to mRNA targets, have been found to bind to CsrA in vitro[20]. In *Bacillus subtilis*, CsrA has been discovered to facilitate base pairing between the SR1 sRNA and its target *ahrC* mRNA[58]. In the case of *P. fuliginea* BSW20308, similar mechanisms may exist between these predicted sRNAs, CsrA and Hfq, especially for those putative trans-encoding sRNA which generally require Hfq for better interaction between the sRNAs and their mRNA targets (Fig. 10). In addition, some cis-encoded sRNAs have also been reported to be able to function in trans. For instance, cis-encoded antisense sRNA SprA1$_{AS}$ was known to regulate its target mRNA SprA1 through imperfect base pairing with the nonoverlapping region[59]. Interestingly, multiple imperfect base-pairing targets were also predicted for those potential cis-encoded sRNAs in *P. fuliginea* BSW20308, in addition to the complementary targets (Supplementary Data 6). This is consistent with previous hypothesis that cis-encoded sRNAs could also regulate imperfect base-paired targets in trans[59]. Since the discovery is based on in silico target prediction, further experimental verification is encouraged to solidify these findings. Moreover, Hfq has been shown to bind to some cis-encoded sRNAs for regulation[60,61]. Therefore, hinted by the discovery in this study and the evidences from the previous studies, complex interaction between sRNAs-CsrA-Hfq merits further investigation (Fig. 10). Future research on this topic will expand the understanding of the functional diversity of the Csr system.

In summary, we have discovered two new Csr sRNAs, Pf2 and Pf3, from the Arctic psychrophilic bacterium *P. fuliginea* BSW20308, and revealed the widespread occurrence of multiple Pf sRNAs across *Pseudoalteromonas* species. The knockout of varying numbers of Pf sRNAs in *P. fuliginea* BSW20308 impacts the physiology and growth of the bacterium, exhibiting a compensatory effect under specific conditions. Furthermore, the CsrA targetomes are influenced by the knockout of Pf sRNAs, showing dynamics and intricacy of the Csr system. This study also detected numerous potential sRNAs within the CsrA targetomes, indicating a complex interplay between CsrA, sRNA, and Hfq, adding further complexity to the Csr system. The results obtained provide valuable insights for future investigations into the diversity in CsrA targets, crosstalk between the Csr system and the Hfq system, regulation mechanisms, and the sRNA world in bacteria living in extreme environments such as the polar regions.

## Materials and methods
### Bacterial strains and plasmids
The strains used in this study are listed in Supplementary Data 7. *E. coli* strains were grown in LB medium (10 g tryptone, 5 g yeast extract, and

10 g NaCl dissolved in 1000 ml of deionized water), supplemented with kanamycin (50 mg/L) or gentamicin (10 mg/L). The WT *P. fuliginea* BSW20308 and its mutation strains were grown in 2216E medium. The plasmids used in this study are listed in Supplementary Data 7, and plasmid construction was performed using the ClonExpress II one step cloning kit (Vazyme Co., Ltd., China), following the manufacturer's instructions.

### Identification and verification of Pf sRNAs
The putative sRNAs were predicted using Rockhopper[62], and manually checked for the presence of GGA motifs. The secondary structures of sRNAs were predicted by Mfold[63]. Putative Csr sRNAs (Pf sRNAs) were further verified following our previous study[24]. For the promoter activity assay, the predicted promoter sequence was cloned into the *Spe*I-digested pRU1701 plasmid upstream of the promoter-less green fluorescent protein (GFP) gene, and then transformed into the *E. coli* DE3 strain. The resulting *E. coli* DE3 strain was cultured overnight in LB medium containing 10 μg/ml gentamicin and then scaled up in flasks for fluorescence measurement. After 6, 9, 12, and 16 h of growth, cells were collected by removing the medium, washed three times, and resuspended in 200 μl of sterile distilled water. The empty plasmid pRU1701 was used as the negative control. Three biological replicates were performed for both the control and experiment groups. The GFP fluorescence intensity was measured using a microplate reader (excitation, 485 nm; emission, 538 nm) and divided by the corresponding $OD_{600}$ value to determine the unit fluorescence intensity.

To confirm the interaction between Pf sRNAs and CsrA in vivo, Pf sRNAs were cloned into pET28a and subsequently transformed into the *E. coli* DE3 strain. Additionally, CsrB and CsrA, both originating from *E. coli*, were cloned into pET28a and transformed into the *E. coli* DE3 strain for comparative purposes. The *E. coli* DE3 strain harboring an empty pET28a plasmid served as the negative control. The *E. coli* DE3 strains carrying pET28a, pET28a-*csrA*, pET28a-*csrB*, pET28a-*pf2*, and pET28a-*pf3* plasmids were inoculated onto Kornberg plates (1.1% $K_2HPO_4$, 0.85% $KH_2PO_4$, and 0.6% yeast extract containing 1% glucose) with 1 mM IPTG (isopropyl-β-D-thiogalactopyranoside) and 30 μg/ml kanamycin. The plates were incubated upright at 37 °C overnight and then stained with iodine for 5 min.

### Pf sRNA gene deletion
The Pf sRNA genes were knocked out using a CRISPR/Cas9-based approach. In brief, 1000-bp upstream and downstream sequences flanking the target Pf sRNAs were cloned into the plasmid pK18-mobsacB-Ery digested with *Bam*HI and *Hind*III. The gRNAs were designed on the online platform (https://zlab.bio/guide-design-resources), using the "Cas-Designer" tool following the instructions of the platform. The gRNAs with the highest scores were selected. The designed gRNA scaffold and Cas9 sequences were then cloned into the pK18 plasmid, digested with *Bgi*I and *Pvu*I, which contained upstream and downstream homologous fragments of the target Pf sRNAs. Expression of the gRNA scaffold and Cas9 protein was enabled by the previously identified D4-12 promoter[24].

The knockout plasmids were transferred into *P. fuliginea* BSW20308 through conjugation, following the established protocols described in previous reports[64]. Initially, the donor strain, *E. coli* WM3064 harboring the knockout plasmids, and the recipient strains were cultured to an $OD_{600}$ value of 0.6–1.0. Subsequently, the donor and the recipient strains were collected by centrifugation at 12,000 rpm for 2 min, and then washed twice with MLB medium (Modified LB medium containing 10 g tryptone, 5 g yeast extract, and 10 g NaCl dissolved in 500 ml deionized water and 500 ml sterile seawater). The combined strain mixture was resuspended in 100 μl of MLB medium and then dropped onto MLB plates supplemented with diaminopimelic acid (DAP). The plates were incubated at 25 °C for 24 h. The bacterial lawn was scraped off, washed multiple times, and then spread onto erythromycin-containing 2216E plates. Single colonies formed at 25 °C were verified through PCR analysis and confirmed by DNA sequencing, ensuring the successful disruption of the target genes.

## Growth measurement at various salinity and sucrose concentrations

The WT, $\Delta Pf1$, $\Delta Pf12$, and $\Delta Pf123$ strains were grown in 2216E medium at 25 °C with 150 rpm overnight for seed preparation. Subsequently, appropriate volumes of the seed cultures were inoculated into 2216E medium under varying conditions to make the start $OD_{600}$ of 0.01. Growth was measured in the 100-well plates (model P-97) by measuring $OD_{600}$ every 0.5 h until the onset of the death phase in the Bioscreen C (Oy Growth Curves Ab Ltd) The tested salinities were 0%, 2%, 4%, 6% and 8% (w/v) while the tested sucrose concentrations were 0%, 2%, 4%, 6%, and 8% (w/v). All experiments were performed in three replicates.

## Construction of CsrA-3×FLAG-expressing strains

The 3×FLAG epitope tag was added at the CsrA C-terminus by homologous recombination in the WT, $\Delta Pf1$, $\Delta Pf12$, and $\Delta Pf123$ strains. In brief, the CsrA protein was fused to a 3×FLAG epitope at its C-terminus by cloning regions encoding ~1000 bp of its C-terminal coding region (C-term) and ~1000 bp downstream of the stop codon (DN) into the plasmid pK18-mobsacB-Ery to flank a 3×FLAG tag. Afterwards, the 3×FLAG-tag constructs were amplified by PCR and transferred into the chromosome of the WT, $\Delta Pf1$, $\Delta Pf12$, and $\Delta Pf123$ strains by conjugation and double-crossover homologous recombination. Briefly, the recombinant plasmid was transformed into *E. coli* WM3064 and co-cultured with recipient strains on the mating plates supplemented with DAP. The plates were incubated overnight at 25 °C. Cells were washed from the plates and resuspended in 200 µl of 2216E medium, then spread onto the 2216E plates with 25 µg/ml erythromycin. Single colonies were picked for PCR analysis to verify the successful fusion of the 3×FLAG tag and confirmed further through DNA sequencing. Growth of the CsrA-3×FLAG-expressing strains were compared with the native CsrA-expressing strains to mitigate any potential growth impact due to the introduction of the 3×FLAG tag.

## SDS-PAGE and western blot analysis

Strains expressing CsrA-3×FLAG fusion proteins were collected by centrifugation at 12,000 rpm for 2 min and re-suspended with PBS buffer. Cells were disrupted via sonication for 5–10 min on ice. Protein loading buffer was added into the cell lysates, and boiled at 100 °C for 5 min followed by SDS-polyacrylamide(PAA) gel electrophoresis (SDS-PAGE). For western blot analysis, samples were transferred from the SDS-PAA gels onto the polyvinylidene difluoride (PVDF) membranes be electroblotting. The PVDF membranes were blocked for 2 h with 50 ml of 5% (w/v) milk powder that dissolved in PBST (PBS with 0.5% Tween-20), and then incubated overnight at 4 °C with primary antibody (Anti-FLAG Tag monoclonal antibody, Sangon Biotech, #D191041, China). Membranes were then washed with PBST solution for three times, followed by incubation with secondary antibody (HRP-conjugated Goat anti-mouse IgM, Sangon Biotech, #D110103, China) for 30 min. After washing with PBST for three times, the blot was developed using High sensitive Plus ECL luminescence reagent (Sangon Biotech, #C520045, China) and visualized using ChemiDoc MP Imaging System (Bio-Rad, USA). The WT, $\Delta Pf1$, $\Delta Pf12$, and $\Delta Pf123$ strains with native CsrA were used as the negative controls.

## RIP-seq

Given that our previous research demonstrated a more significant induction of differentially expressed genes at 32 °C compared to low temperatures[24], the current study focused on assessing the impact of varying Pf sRNA knockouts at 32 °C. RIP-seq experiments were performed with three biological replicates. The CsrA-3×FLAG-expressing strains were grown to the late-exponential phase in 2216E medium at 32 °C. Cells were then centrifugated and washed twice with pre-cooled PBS buffer. The washed cells were resuspended in PBS buffer containing 0.3% formaldehyde and then shaken for 10 min at room temperature for crosslinking. The quenching buffer, containing a final concentration of 0.2 M glycine, was added and shaken for 5 min at room temperature. Cells were then centrifuged, washed twice with PBS buffer, and lysed with RIP buffer (25 mM Tris, pH 7.4,

150 mM KCl, 5 mM EDTA, 0.2% CA-630, and 0.05% SDS) on ice for 1 h, followed by sonication. The lysates were centrifuged to remove cell debris and a portion of the supernatants were saved as input samples, which served as a blank control to mitigate background noise and peak errors. The remaining samples were then incubated with prewashed anti-FLAG M2 paramagnetic beads (Sigma) at 4 °C overnight with gentle mixing. The bead-FLAG-CsrA-RNA complexes were washed multiple times in NT2 buffer (50 mM Tris-HCl pH 7.5, 150 mM NaCl, 1 mM MgCl₂, 0.05% NP-40 and protease inhibitor mixture). The washed complexes were then treated with DNase I at 37 °C for 15 min, and further treated with proteinase K at 55 °C for 30 min with shaking. Afterwards, total RNA was extracted with phenol:chloroform:isoamyl alcohol (25:24:1, v/v/v) and dissolved in RNase-free water. Both RNAs from the input samples and immunoprecipitation (IP) samples were used for library construction and high throughput sequencing. Libraries were constructed with NEB Next® Ultra™ RNA Library Prep Kit for Illumina (NEB) and sequenced on Nova Seq 6000 (Illumina) by DIA-TRE Biotechnology (Shanghai, China).

## Sequence data analysis

The quality of raw data generated by RIP-seq was checked using FastQC version 0.11.9 (https://www.bioinformatics.babraham.ac.uk/projects/fastqc). Adapters and low-quality sequences were trimmed using Cutadapt version 1.18[65]. Clean reads were mapped to the *P. fuliginea* BSW20308 genome (CP013138.1 and CP013139.1) using STAR version 2.7.1a[66]. Alignment SAM files were sorted by Samtools version 1.3.1[67]. The RIP-seq peaks were called by MACS2, with the "narrow" peak calling option and a q value threshold of 0.01[68]. Peak calling was performed independently against input samples for each replicate, and the peak regions identified in all replicates were merged to define the peaks for differential binding analyses. For motif analysis and annotation, all peaks in the respective samples were chosen and processed using HOMER 4.1.5[69]. The peak sequences served as the input, while a randomly chosen genomic region of *P. fuliginea* BSW20308 was used as the background. For motif enrichment analysis, ZOOPS scoring and binomial distribution were utilized by HOMER and MEME[70]. Motif drawing was achieved using the R package motifStack. The length of the motifs was set to 8, 10, and 12 base pairs. The CsrA interactome network diagram was generated using Cytoscape 3.10.1[71].

## Electrophoretic gel mobility shift assays (EMSA)

The binding of CsrA to RNA was determined through EMSA following previous studies[7,72]. The *csrA* gene from *P. fuliginea* BSW20308 was cloned into the pET28a plasmid and expressed as a recombinant protein with a C-terminal His tag in *E. coli* DE3. The resulting recombinant CsrA-His₆ protein was purified using affinity chromatography. RNAs for EMSA were synthesized in vitro using the T7 High Yield RNA Synthesis Kit (Sangon Biotech, China). The binding reactions were set up with 0.1–0.5 nM RNA, 40 mM NaCl, 4 mM Tris, 4 mM MgCl₂ (pH 8.0), 4% glycerol (W/V), 100 mg/ml BSA (non-specific protein control), 20 mM DTT and the purified CsrA-His₆ protein. The incubation was performed at room temperature for 30 min. The reaction mixtures were separated on a native PAA gel. The gels were stained with NA-red (Beyotime, China) and visualized using the Gel Doc™EZ imager (Bio-Rad, USA). For competitive EMSA, target mRNAs were labeled with Fluorescein-12-UTP (Beyotime, China) during in vitro transcription. In the competitive binding reactions, unlabeled Pf sRNA was first incubated with the purified CsrA-His₆ protein at room temperature for 30 min to allow pre-binding. Subsequently, Fluorescein-labeled target mRNA was added to the reaction mixture, followed by another 30 min incubation. The final reaction mixtures were separated on a native PAA gel under the same conditions as described above. Fluorescent signals of the labeled mRNA were visualized using the Gel Doc™ EZ imager (Bio-Rad, USA), allowing the observation of competitive displacement by unlabeled RNA.

## Cell motility and biofilm assays

Strains were cultured to the exponential growth phase ($OD_{600}$ of 0.4), centrifuged, and resuspended in fresh 2216E medium for motility and

biofilm assays. Cell motility was measured using semisolid agar plates in triplicate. For each strain, 5 μl of re-suspended cells were placed in the center of a 2216E plate containing 0.3% agar and incubated at 32 °C for over 24 h. The colony diameter was measured. To minimize variability between plates, we also performed motility assays on a single 2216E plate containing 0.3% agar, where 5 μl of resuspended cells from all four strains were inoculated in separate, equidistant positions on the same plate.

Biofilm assay was modified from previous studies[73]. In brief, 300 μl of resuspended cells were added to each well of a 96-well plate, with five replicates for each strain and 2216E medium as a blank control. After incubation at 32 °C for over 24 h, the liquid culture was carefully discarded, and the plate was rinsed three times with sterile water. Then 300 μl of icy acetone was added to each well to fix the biofilm. After removing the acetone, 300 μl of crystal violet was added to each well and stained for 20 min, followed by gentle rinsing three times with sterile water. Subsequently, 300 μl of acetic acid was added to solubilize the crystal violet staining agent and allowed to stand for 30 min. Finally, using 300 μl of acetic acid as a blank, the $OD_{570}$ of the samples was measured using a microplate reader.

## Structure prediction
AlphaFold 3 was used to predict the potential interaction between CsrA, Hfq, and sRNAs. Sequences were submitted to the web server (https://alphafoldserver.com/)[74]. Structures were visualized and edited in PyMOL 2.5.4[75].

## Statistics and reproducibility
Statistical analyses were conducted using GraphPad Prism version 9 (GraphPad Software). Student's $t$-test was applied to compare two groups, while one-way ANOVA was used for multiple-group comparisons. Results are expressed as mean ± s.e.m., and statistical significance was defined as $P < 0.05$. To ensure reproducibility, all experiments were independently repeated at least three times, with consistent findings across replicates. For experiments without formal statistical analysis, reproducibility was assessed by conducting repeated experiments under the same conditions to verify consistency in results.

## Reporting summary
Further information on research design is available in the Nature Portfolio Reporting Summary linked to this article.

## Data availability
All data produced or analyzed in this study are provided in the article and its supplementary information files. Source data can be found in Supplementary Data 8. The raw RIP-seq data reported in this study has been uploaded in NCBI's Sequence Read Archive (SRA, accession numbers: SRR29791929 to SRR29791952) under the BioProject accession number: PRJNA1134920.

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

## Acknowledgements

The authors thank Professor Ben Luisi from the University of Cambridge for his invaluable insights and constructive suggestions throughout the writing process of this paper and structural analysis, as well as Professor Xiulan Chen from Shandong University for providing pK18-mobsacB-Ery plasmid. The project was supported by the National Key Research and Development Program of China (Grant No. 2022YFC2807501) and the National Natural Science Foundation of China (Grant No. 42476264 and 41976224).

## Author contributions

Zedong Duan performed most of the experiments and data analysis and drafted the paper. Li Liao supervised the study and revised the paper. Tingyi Lai, Ruyi Yang, and Jin Zhang conducted part of experiments. Bo Chen supervised the study.

## Competing interests

The authors declare no competing interests.
