## [Transparent Peer Review file · Communications Biology]

Dynamic and intricate regulation by the Csr sRNAs in the Arctic *Pseudoalteromonas fuliginea*

Corresponding Author: Professor li liao

Version 0:

Reviewer comments:

Reviewer #1

(Remarks to the Author)

The Csr system is pivotal in controlling various physiological processes via the interaction of its antagonistic sRNAs to interfere its recognition of its mRNA targets. The authors reported two new sRNAs of the Csr system and studied the impact of these Pf sRNAs on CsrA targetomes in the Arctic bacterium *Pseudoalteromonas fuliginea* BSW20308. The authors analyzed the RIP-seq of CsrA-targetome of the wild type and the Pf sRNA mutants in depth and revealed CsrA system had a significant influence on various aspects. However, this manuscript lacks some conclusions in all of the analysis of the RIP-seq, which makes it hard to get the information of which and how Pf sRNAs influence the Csr system. Here are the specific comments.

1. Fig. 2, the growths of Pf12 and Pf123 mutants are undistinguishable in 2216E medium and it indicates that Pf2 is predominant in normal 2216E medium. Thus, the conclusion in line 130-131 is not proper.
2. Line 179-185, it is just a rough description of mapped KEGG, which do not give us any information about the influence of Pf sRNA knockout on the CsrA targets. The authors should analyze further and pinpoint the differences between WT and the Pf mutants.
3. In each part of CsrA modulates certain cellular process, as mentioned above, the authors should choose at least one target gene revealed by RIP-seq and conduct experimental verifications, such as EMSAs. And also, the competing EMSAs should be performed to show the addition of sRNA indeed interferes the interaction between CsrA and its target mRNA.
4. Line 262-269, what about the WP_007377877? Does CsrA interact with it?
5. Line 285-286, the genes encoding c-di-GMP means the genes involved in its biosynthesis? This should be clarified.

Reviewer #2

(Remarks to the Author)

The carbon storage regulator system is present in numerous bacterial species, where they show a high degree of functional diversity. Commonly, the system consists of several small non-coding regulatory RNAs that efficiently sponge away the protein CsrA from its mRNA-binding sites. In previous studies, the authors identified this system in the species *Pseudoalteromonas fuliginea* and started some initial characterization. In this submitted work presented here, they further identified two more sRNAs, Pf2 and Pf3, and show that both are components of the Csr regulatory system. Additive (but unfortunately no specific) mutants exhibit an additive phenotype on cell growth. The authors then performed RIPseq analysis with FLAG-tagged CsrA in the wild type and the various Pf1-3 mutants. By this, identified a huge amount of potential CsrA targets. While there is no in further characterization of the targets in any depth (unfortunately), the authors provide some evidence that the Csr system affects complex processes such as flagella-mediated motility or biofilm formation. No further evidence was provided for also identified potential regulation of cell envelope processes or type 6 secretion system formation and activity. In addition, the RIPseq identified a number of regulatory RNAs, suggesting a significant overlap with Hfq-based regulation. However, also this was not backed up by actual experiments.

The authors provide an interesting start for a number of up-following studies. It's a shame that the very interesting findings are not really followed up already, at least to some extent. The authors missed a huge chance by not using single Pf1, Pf2 and Pf3 mutants to determine potential specificities (which is suggested by some of the experiments). It would have also been great if the relative amounts of Pf1-3 would have been determined.

Further issues are listed below:

Major:

Paragraph starting at line 119/line 127: The different responses of the mutants point at different regulation and/or roles of the Pf sRNAs. Since there is no comparative analysis of the corresponding sRNA levels, it is also not that easy to predict if it's just the abundance of the RNAs and the number of potential CsrA-binding sites. This is quite a missed chance here, I would have expected at least the single deletions (if not all the different combinations).
A *csrA* mutant would be good as a control.

Line 184: It may have been more interesting if the authors would have gone after the unknown functions of Csr – i.e., a function in flagella-mediated motility is long known.

In general: Is the complete lack of CsrA antagonist RNAs a realistic physiological scenario? Is it possible that in the complete absence of Pf1-3 there is more (and unspecific) binding of CsrA?

Paragraph starting line 215: Can the authors provide any somewhat specific phenotypes underlying their observation of potential regulation?
The same goes for the T6SS – are there any phenotypes the authors can provide in the mutants, such as the presence of the corresponding proteins?
Again the same for the biofilms, at least some key players may have been looked into.

Line 320: Hfq, its role and properties need to be explained before.

Minor

Line 86: between which genes are the Pf2 and Pf3 genes located?
Line 87: Out of how many chromosomes? Omit 'up to'.
Line 93: maybe add 'predicted' or 'putative' before promoters
Figure 1c: It would be good if the OD curves could be added to see whether the promoter activity is correlating to specific growth phases.
Line 95: add 'heterologously in *E. coli*'
Line 108: is it the same or a different genetic context?
Supplementary Fig.1/Legend: What are the different lanes?
Line 155: please mention already here that the FLAG-tagged CsrA is (fully) active. Please show the control.
Line 181: two-component signal transduction or general signal transduction (or both), please clarify.
Line 213; Materials and Methods: I believe the results, still the strains to be compared should be on the same plate as there are always significant differences between the properties of the soft-agar plates.
Line 278: please add 'static' to the biofilm
Line 301: was detected.
Line 304: the GGA motif
Line 324: dimers

Reviewer #3

(Remarks to the Author)

The carbon storage regulator (Csr) system is widely distributed across most bacteria and plays a crucial role in regulating various cellular processes. This manuscript reported two newly identified Csr sRNAs, Pf2 and Pf3, in the Arctic bacterium *Pseudoalteromonas fuliginea* BSW20308. The authors further investigated the functional mechanisms of these sRNAs and found that both sRNAs modulate the interactions between the global regulator CsrA and target RNAs, thereby regulating several cellular pathways, including type VI secretion system, cell motility, and cell envelope formation.

Minor Comments

1. The authors constructed Δ Pf1, Δ Pf12, Δ Pf123 and WT strains for RIP-Seq analysis. Please provide an explanation for the reason not to construct Δ Pf2, Δ Pf3, and Δ Pf23 strains.
2. In Fig. 10, the authors showed two types of sRNAs, "sponge sRNAs" and "base-pairing sRNAs", which are colored in red and green, respectively. Please add this color coding to the figure caption. Additionally, in the right upper panel, which shows the involved cellular pathways, please clarify the square and round shapes represent?

Version 1:

Reviewer comments:

Reviewer #1

(Remarks to the Author)

All my concerns have been addressed.

Reviewer #2

(Remarks to the Author)

Thanks to the authors for addressing my concerns adequately.

Dear Editor and Reviewers,

We sincerely thank the Editor and Reviewers for your time and effort in handling our manuscript titled “Dynamic and intricate regulation by the carbon storage regulation system in the Arctic *Pseudoalteromonas fuliginea*” (reference number: COMMSBIO-24-6090-T) and for providing us with the opportunity to further refine and enhance our work. We greatly appreciate the thoughtful consideration of our submission and the constructive feedback shared by the reviewers. Your insightful comments and suggestions have been invaluable in enhancing the quality and clarity of our manuscript, and we are truly grateful for the inspiration and guidance you have provided throughout this process.

In response to the meticulous comments provided by the reviewers, we have undertaken comprehensive revisions that encompass structural adjustments, enriched data analyses, and addressed concerns regarding experimental validation through the execution of additional experiments. With the substantial amount of new data and revisions, we believe our manuscript has improved significantly. We sincerely hope it now aligns with the rigorous standards upheld by Communications Biology.

Below are our point-by-point responses to the reviewers' valuable comments, including the exact location where the change can be found in the revised manuscript. For your convenience, we use black font for the reviewers' comments, blue for our responses, and red for the changes in the revised manuscript.

Reviewer #1 (Remarks to the Author):

The Csr system is pivotal in controlling various physiological processes via the interaction of its antagonistic sRNAs to interfere its recognition of its mRNA targets. The authors reported two new sRNAs of the Csr system and studied the impact of these Pf sRNAs on CsrA targetomes in the Arctic bacterium *Pseudoalteromonas fuliginea* BSW20308. The authors analyzed the RIP-seq of CsrA-targetome of the wild type and the Pf sRNA mutants in depth and revealed CsrA system had a significant influence on various aspects. However, this manuscript lacks some conclusions in all of the analysis of the RIP-seq, which makes it hard to get the information of which and how Pf sRNAs influence the Csr system. Here are the specific comments.

Response:

We express our heartfelt gratitude for recognizing our research endeavors and for providing us with insightful and invaluable feedback. We acknowledge your valid point regarding the absence of conclusions in our RIP-seq data analysis. In response to your concern and to enhance the comprehensibility of our findings, we have revised the manuscript to include a concluding paragraph at the end of the Discussion section (please see the lines 483-493 in the revision). In this new paragraph, we have succinctly summarized the pivotal insights derived from the RIP-seq data and have explored the potential interconnections among the three Pf sRNAs. With these amendments, we sincerely hope that we have met your expectations and further enriched the manuscript's overall quality.

1. Fig. 2, the growths of Pf12 and Pf123 mutants are undistinguishable in 2216E medium

and it indicates that Pf2 is predominant in normal 2216E medium. Thus, the conclusion in line 130-131 is not proper.

Response:

We appreciate the reviewer pointing this out. Indeed, as we reported in the manuscript, the growth of both $\Delta Pf12$ and $\Delta Pf123$ mutants is quite similar in normal 2216E medium, yet both exhibit inferior growth compared to the wild type and $\Delta Pf1$. This observation led us to infer that the impact on growth becomes more pronounced with the deletion of additional Pf sRNAs. However, as the reviewer correctly noted, this does not rule out the possibility that Pf2 is the predominant factor, based on our previous data.

To determine whether Pf2 or the number of deletions is more influential, we have constructed two additional mutants: $\Delta Pf2$ and $\Delta Pf3$. Growth was measured under the same conditions as before. The results (please see the figure below) indicate that the growth of the three single deletion mutants is indistinguishable. These findings suggest that neither Pf1, Pf2, nor Pf3 alone significantly impacts cellular growth. Consequently, this does not support the conclusion that Pf2 is the predominant player. Instead, it supports our previous hypothesis that single deletions may not severely affect growth, but the impact increases with the deletion of more Pf sRNAs.

However, since no apparent difference in growth was observed between $\Delta Pf12$ and $\Delta Pf123$ in normal 2216E (although triple deletion $\Delta Pf123$ showed apparently worse growth than double deletion $\Delta Pf12$ in 2216E medium plus 4% sucrose), it is improper to conclude that only the number of deletions matters. It appears that the influence depends not only on the number of deletions but also on the specific conditions tested, which merits further research in our future studies but is not the focus of the current manuscript.

Therefore, to avoid any misleading interpretation, we have revised the conclusion as follows (lines 137-144 in the revision):

Notably, the growth of the double ($\Delta Pf12$) and triple ($\Delta Pf123$) deletion mutants was severely compromised (Fig. 2a). Although the growth curves of $\Delta Pf12$ and $\Delta Pf123$ were nearly identical in standard 2216E media and in 2216E supplemented with 4% NaCl, the growth of $\Delta Pf123$ was notably inferior to that of $\Delta Pf12$ in 2216E media containing 4% sucrose. This finding generally aligns with the notion that these Pf sRNAs exhibit some degree of functional complementarity, although their performance may vary under specific conditions.

Supplementary Fig. 2. Growth curves of Pf sRNA mutants and the wild type. Growth curves of the wild type (WT) and single sRNA mutant strains ($\Delta Pf1$, $\Delta Pf2$, $\Delta Pf3$) were monitored over time.

2. Line 179-185, it is just a rough description of mapped KEGG, which do not give us any information about the influence of Pf sRNA knockout on the CsrA targets. The authors should analyze further and pinpoint the differences between WT and the Pf mutants.

Response:

We appreciate your insightful comments. To address your concern, we further analyzed the enrichment of genes in KEGG pathways. This analysis revealed notable differences between the wild-type (WT) and the Pf sRNA mutants. Specifically, the number of significantly enriched pathways varied among the strains, with 28 pathways enriched in WT, 25 in $\Delta Pf1$, 17 in $\Delta Pf12$, and 20 in $\Delta Pf123$. This indicates that the deletion of sRNAs does affect the CsrA targetomes. For example, certain pathways enriched in WT, such as DNA replication and repair, and propanoate metabolism, were not enriched in the sRNA mutant strains. Moreover, even for pathways that were consistently enriched across WT and the mutants, such as those related to cell motility, two-component systems, signal transduction, carbohydrate metabolism, and secretion systems, we observed differences in the number and types of genes contributing to these pathways. This highlights that sRNA deletions can alter the extent to which specific genes within these pathways are regulated by CsrA.

These findings have been included in the revised manuscript to provide a more detailed and precise description of the impact of Pf sRNA knockouts on the CsrA targetomes. Thank you for pointing out this important aspect, which has enabled us to present a clearer and more comprehensive analysis.

We have also outlined the specific changes we made to the manuscript below for your convenience:

Lines 195-206 in the revision. “The enrichment of genes in KEGG pathways corresponding to all CsrA RIP-seq peaks was systematically analyzed (Supplementary Fig. 5). The knockout of sRNAs affected the CsrA targetomes in two ways: by changing the enriched categories of KEGG pathways and by altering the level of involvement of specific genes within the same pathways. Specifically, fewer pathways were enriched in the mutants compared to the wild strain (17 to 25 in the mutants versus 28 in the wild strain). Notably, some pathways such as DNA replication and repair and propanoate metabolism were exclusively enriched in the wild strain. Furthermore, even for pathways that were enriched across all strains, there were variations in the number and types of genes contributing to specific pathways. Detailed information on the enriched functions of genes with CsrA RIP-seq peaks and complete pathway results were shown in the Supplementary Data 3. Several of these enriched functional pathways corresponding to the known functions of CsrA and potential new targets were further analyzed in subsequent sections.”

3. In each part of CsrA modulates certain cellular process, as mentioned above, the authors should choose at least one target gene revealed by RIP-seq and conduct experimental

verifications, such as EMSAs. And also, the competing EMSAs should be performed to show the addition of sRNA indeed interferes the interaction between CsrA and its target mRNA.

Response:

We sincerely appreciate your thoughtful comments and constructive suggestions. In response to your recommendation, we have performed a series of electrophoretic mobility shift assays (EMSAs) and competing EMSAs to experimentally validate the interactions between CsrA and its target mRNAs, as well as the modulatory role of sRNAs.

For this purpose, we have selected *fleQ* (cell motility), *pal* (cell envelope), *ompR* (cell envelope and biofilm), *h-ns* (biofilm), WP_033023891 and WP_007377080 (T6SS, validated in our original manuscript) as representative target genes, each corresponding to a distinct cellular process identified through our RIP-seq analysis. These genes have been validated by EMSA during this revision, while WP_033023891 and WP_007377080 were validated in our original submission. The EMSA results confirmed that these target genes can bind to the CsrA protein *in vitro* (figures are shown below). Furthermore, we have conducted competitive EMSA experiments for genes encoding WP_033023891 and WP_007377080 (both belong to T6SS), *pal* (cell envelope), and *ompR* (cell envelope and biofilm). The results show that the binding of CsrA to its target mRNAs is disrupted after the introduction of Pf sRNAs. These experiments demonstrate that the addition of sRNAs effectively interferes with the interaction between CsrA and its target mRNAs, thereby supporting the regulatory mechanism we proposed.

The detailed results and analyses have been incorporated into the revised manuscript. Specifically, the EMSA results for *fleQ* have been added to Fig. 4d, those for *pal* have been included in the Fig. 5, and the EMSA results for *h-ns* and *ompR* have been incorporated into Fig. 7c. Additionally, the results of the competitive EMSA experiments are now presented in the Supplementary Fig 8. We sincerely appreciate your insightful feedback, which has significantly enhanced the depth and rigor of our study.

We have presented the EMSA results separately below for your convenience:

Above are the electrophoretic mobility shift assays (EMSAs) to assess the binding of CsrA to *fleQ* (cell motility), *pal* (cell envelope), *ompR* (cell envelope and biofilm), and *h-ns* (biofilm) mRNAs. The results demonstrate clear shifts in the presence of CsrA, confirming specific interactions between CsrA and these mRNAs, whereas no such shifts are observed with BSA (bovine serum albumin, the negative control). These findings highlight the specific binding of CsrA and its target mRNAs.

Below are the competitive EMSA results (supplementary Fig. 8). The results demonstrate that all three Pf sRNAs reduce the binding of CsrA to its target mRNAs to varying degrees, suggesting their roles as modulators of CsrA-mRNA interactions.

Supplementary Fig. 8. Pf sRNAs interfered with the interaction between CsrA and its target mRNAs. Electrophoretic mobility shift assays (EMSAs) were performed to assess the effects of unlabeled Pf sRNAs (Pf1, Pf2, and Pf3) on the binding of CsrA to its target mRNAs (WP_033023891, ompR, pal, and WP_007377080). The presence (+) or absence (-) of unlabeled Pf sRNAs and CsrA is indicated above each panel.

4. Line 262-269, what about the WP_007377877? Does CsrA interact with it?

Response:

We sincerely appreciate your insightful question regarding WP_007377877 and its potential interaction with CsrA. We conducted an electrophoretic mobility shift assay (EMSA) and confirmed that the mRNA of WP_007377877 can indeed bind to the CsrA protein *in vitro*.

As noted in our manuscript, the specific function of this protein remains unclear, and its role in relation to the type VI secretion system requires further investigation. We acknowledge the importance of understanding this connection and aim to explore it in future studies to provide a more comprehensive insight into its biological significance. We have included the additional data in the revised manuscript to address your concerns.

We have also outlined the specific changes we made to the manuscript below for your convenience:

Lines 294-296 in the revision. “As for WP_007377877, a potential CsrA binding site was identified within its coding region, and EMSA results demonstrated that the mRNA corresponding to this region could bind to CsrA *in vitro* (Fig. 6c, d).”

Figure 6. The Type VI Secretion System and its differential regulation across strains. (a) Schematic representation of the T6SS gene cluster in *P. fuliginea* BSW20308. Genes are depicted as arrows pointing in the direction of transcription. The gene names are marked above. White indicates unknown genes. The term 'immunity' refers to genes encoding for putative immunity proteins that neutralize T6SS effectors. (b) The structural organization of T6SS and the related genes detected in the CsrA targetomes across strains. Squares near gene names show the detection in the WT (left), $\Delta Pf1$ (top), $\Delta Pf12$ (right), and $\Delta Pf123$ (bottom) strains, with green representing a positive result and white representing a negative result. (c) Predicted CsrA-mRNA binding sites for mRNAs of WP_033023891, WP_007377080 and WP_007377877. Putative CsrA binding sites are marked with blue circles, while GGA motifs involved in binding are indicated with purple arcs. Red squares mark the location of start codons. (d) EMSA of 1 μ M CsrA binding to mRNAs of WP_033023891, WP_007377080 and WP_007377877, with 1 μ M BSA as the control.

5. Line 285-286, the genes encoding c-di-GMP means the genes involved in its biosynthesis? This should be clarified.

Response:

Thank you for your comment. We apologize for the ambiguity of our description. The gene (D172_RS02615) encodes diguanylate cyclase (DGC), which synthesizes c-di-GMP from two molecules of guanosine triphosphate (GTP). We have modified the relevant description as follows:

Lines 312-313 in the revision. “Indeed, genes encoding sigma factors RpoE and RpoS, and the diguanylate cyclase that synthesizes c-di-GMP, were detected in the CsrA targetomes”

Reviewer #2 (Remarks to the Author):

The carbon storage regulator system is present in numerous bacterial species, where they show a high degree of functional diversity. Commonly, the system consists of several small non-coding regulatory RNAs that efficiently sponge away the protein CsrA from its mRNA-binding sites. In previous studies, the authors identified this system in the species *Pseudoalteromonas fuliginea* and started some initial characterization. In this submitted work presented here, they further identified two more sRNAs, Pf2 and Pf3, and show that both are components of the Csr regulatory system. Additive (but unfortunately no specific) mutants exhibit an additive phenotype on cell growth. The authors then performed RIPseq analysis with FLAG-tagged CsrA in the wild type and the various Pf1-3 mutants. By this, identified a huge amount of potential CsrA targets. While there is no in further characterization of the targets in any depth (unfortunately), the authors provide some evidence that the Csr system affects complex processes such as flagella-mediated motility or biofilm formation. No further evidence was provided for also identified potential regulation of cell envelope processes or type 6 secretion system formation and activity. In addition, the RIPseq identified a number of regulatory RNAs, suggesting a significant overlap with Hfq-based regulation. However, also this was not backed up by actual experiments.

The authors provide an interesting start for a number of up-following studies. It's a shame that the very interesting findings are not really followed up already, at least to some extent. The authors missed a huge chance by not using single Pf1, Pf2 and Pf3 mutants to determine potential specificities (which is suggested by some of the experiments). It would have also been great if the relative amounts of Pf1-3 would have been determined.

Response:

We are deeply grateful for your insightful and meticulous feedback, which has significantly enhanced both the quality and depth of our research. To address your concerns, we have conducted meticulous additional experiments and analyses. Firstly, we have performed EMSA and competitive EMSA experiments on representative targets revealed by RIP-seq. Specifically, targets involved in flagellar-mediated motility (*fleQ*), *pal* (cell envelope), *ompR* (cell envelope and biofilm), and *h-ns* (biofilm) have been verified

through EMSA, and WP_033023891 (The Type VI Secretion System), WP_007377080 (The Type VI Secretion System), *pal* (cell envelope), and *ompR* (cell envelope and biofilm) have been verified through competitive EMSA. All the results (please also check our response to reviewer #1, comment #3) apparently demonstrated specific binding of target mRNAs with CsrA and competitive interference by Pf1, Pf2 and Pf3 sRNAs. These data further support the validation of RIP-seq data. The additional data and analysis have been added in the revision (Please check lines 235, 243 and 316).

Secondly, as to the potential regulatory RNA targets identified by RIP-seq, it indicates cross talk between CsrA- and Hfq-mediated regulation systems, which is very intriguing. However, the verification is much trickier than the protein-encoding mRNA targets due to the uncertainty of the predicted regulatory sRNAs. Although we have done additional verification of the binding of a putative sRNA by CsrA (Please check line 347), which bolsters confidence in our RIP-seq data, we believe that further verification and identification of those predicted regulatory sRNAs should be the first step. This process will require significant additional work and will be a focus of our next paper. Therefore, in this manuscript, we opt to concentrate on highlighting the potential and posing intriguing questions regarding the cross-talk between CsrA- and Hfq-mediated regulation systems, thereby maintaining the focus of our current work.

Lastly, to address your question about not using single Pf1, Pf2 and Pf3 mutants to determine potential specificities, we have constructed and analyzed individual deletions of *Pf1*, *Pf2*, and *Pf3* to assess their individual contributions to cell growth (please check the figure below). Our results demonstrated that the deletion of any single sRNA has minimal to no impact on growth. However, the deletion of multiple Pf sRNAs exerted a significant impact on growth (as shown in Fig.2a). Therefore, we are interested in understanding the effect of the presence of different numbers of Pf sRNAs on the CsrA interactomes as our first step, as described in our manuscript: "The primary objective of this study was to investigate the impact on the CsrA interactome and, subsequently, growth behavior in relation to varying numbers of Csr sRNAs. Consequently, we refrained from constructing all possible mutation combinations of Pf sRNAs." Nevertheless, your idea about comparing the impact of the single deletion of Pf1, Pf2 and Pf3 on CsrA targetomes is of interest, and we would like to further pursue this topic in our future work.

We greatly appreciate your valuable feedback, which has allowed us to further strengthen our study and clarify the significance of our findings. We hope the revised manuscript addresses your concerns and provides a clearer understanding of the roles of *Pf1*, *Pf2*, and *Pf3* in the Csr regulatory network.

Major:

Paragraph starting at line 119/line 127: The different responses of the mutants point at different regulation and/or roles of the Pf sRNAs. Since there is no comparative analysis of the corresponding sRNA levels, it is also not that easy to predict if it's just the abundance of the RNAs and the number of potential CsrA-binding sites. This is quite a missed chance here, I would have expected at least the single deletions (if not all the different

combinations).

A *csrA* mutant would be good as a control.

Response:

Thank you very much for your thoughtful and constructive comments. We think the first part of this comment is similar to the above general comment about the other two single deletions ($\Delta Pf2$ and $\Delta Pf3$). Please refer to our response above.

In response to your suggestion, we have constructed the $\Delta Pf2$ and $\Delta Pf3$ single-sRNA mutant strains and measured their growth curves. We found that the growth curves of $\Delta Pf2$ and $\Delta Pf3$ were similar to that of the wild-type (WT), with no significant growth defects observed. These findings have been incorporated into the revised manuscript as Supplementary Figure 2. However, double and triple deletions of Pf sRNAs showed apparent impact on growth. Therefore, in this manuscript, our primary objective is to compare the targetomes across varying numbers of sRNA deletions.

Regarding the observed phenomenon, we hypothesize that single deletions of Pf sRNAs likely result in relatively minor alterations to the CsrA targetomes compared to multiple deletions, due to the compensatory effect among different sRNAs. Indeed, according to our new qPCR data, upon the single deletion of one sRNA, the remaining sRNAs, particularly Pf1 and Pf2, can partially compensate for the loss by increasing their expression levels (please see the figure below). This compensatory mechanism is further reflected in the growth curves of the single deletion mutants, which show similar trends, suggesting that single deletions may not disrupt the overall regulatory network to the extent seen in double or triple deletions. This observation implies that the functional overlap among the sRNAs might mitigate the effects of single deletions. However, we do not entirely rule out the possibility that each sRNA may regulate a distinct set of targets under specific conditions, meriting further investigation in our future endeavors.

Above is the the relative expression levels of sRNAs Pf1, Pf2, and Pf3 in $\Delta Pf1$, $\Delta Pf2$, and $\Delta Pf3$ mutant strains, measured relative to the WT.

Thank you very much for your insightful suggestion regarding the inclusion of a $\Delta csrA$ mutant as a control. We fully agree that a *csrA* knockout strain would provide valuable insights into the functional roles of the Csr system. During the course of our study on the functions of Pf sRNAs, we attempted to construct a *csrA* knockout strain in

Pseudoalteromonas fuliginea. However, all attempts to generate a viable *csrA* deletion mutant were unsuccessful. This outcome is consistent with previous reports in other bacterial species, such as *Escherichia coli*, *Vibrio cholerae*, and *Legionella pneumophila*, where CsrA has been shown to be an essential gene [Timmermans J, Van Melderen L. Conditional essentiality of the *csrA* gene in *Escherichia coli*. *J Bacteriol* 191, 1722-1724 (2009); Mey AR, Butz HA, Payne SM. *Vibrio cholerae* CsrA Regulates ToxR Levels in Response to Amino Acids and Is Essential for Virulence. *6*, 10.1128/mbio.01064-01015 (2015); Molofsky AB, Swanson MS. *Legionella pneumophila* CsrA is a pivotal repressor of transmission traits and activator of replication. *Mol Microbiol* 50, 445-461 (2003)]. While we recognize the value of a *csrA* mutant for this study, the essential nature of the *csrA* gene in *P. fuliginea* precluded its inclusion in our experiments. We were unable to include a *csrA* knockout strain as a direct control in our study.

We are deeply grateful for your critical insights and constructive suggestions, which have significantly advanced our understanding of the functions of these sRNAs. We have incorporated the experimental results and analyses mentioned above into the manuscript and revised the conclusion section accordingly to ensure the accuracy and completeness of the paper.

Line 184: It may have been more interesting if the authors would have gone after the unknown functions of Csr – i.e., a function in flagella-mediated motility is long known.

Response:

Thank you for your insightful comment regarding the exploration of unknown functions of the Csr system. We completely agree that this represents an exciting and important research direction.

In alignment with your suggestion, we have added a dedicated section in the revised manuscript to describe the potential specific targets of CsrA identified in our study. In this new section (the section is shown below), we have provided detailed insights into these targets and their possible regulatory roles. Additionally, we have listed these specific targets in the Supplementary Data 5 for further clarity. Your suggestion has inspired us to consider extending our work in this direction, and we hope to delve into these aspects in our follow-up studies.

While investigating novel functions is undoubtedly an important direction, we would like to highlight that even within the scope of its known functions, there remain many uncharacterized mRNA targets and previously underexplored regulatory mechanisms. For instance, our study identified several novel mRNA targets related to T6SS, a critical mechanism for bacterial competition and survival. While the involvement of the Csr system in T6SS regulation has been previously reported, the mechanistic insights into how CsrA and its associated sRNAs influence T6SS remain limited. T6SS plays a crucial role in microbial competition by enabling bacteria to inject effector proteins into neighboring cells or the surrounding environment. This system not only aids in microbial survival but also helps shape bacterial community structures by outcompeting rival species. Moreover, most *Pseudoalteromonas* genomes harbor conserved genes encoding structural components of T6SS, yet detailed studies on T6SS functions and regulation in this genus are lacking. Our findings suggest that Pf sRNAs play an important role in modulating CsrA's impact on T6SS-related genes, which may provide new insights into the ecological significance of

T6SS in marine bacteria, especially in marine environments.

Furthermore, the primary aim of our study was to investigate the regulatory roles of newly identified Pf sRNAs on the CsrA targetome and their effects on key cellular pathways. While we recognize the value of further exploring the unknown functions of CsrA, our study provides a foundational understanding of how Pf sRNAs influence CsrA-mediated regulation, which may pave the way for future studies into CsrA's broader functional repertoire. We sincerely appreciate your thoughtful feedback, as it has helped us identify valuable opportunities for further exploration.

We have also outlined the specific changes we made to the manuscript below for your convenience:

Lines 324-337 in the revision. To identify specific targets of the Csr system in *P. fuliginea* BSW20308, we analyzed 616 gene targets detected through RIP-Seq (Supplementary Data 5), of which 138 were classified as specific targets compared to *E. coli*. Among these, several represent previously unreported targets, including D172_RS01480 (*nemA*), involved in the degradation of toxic compounds; D172_RS00585 (cytochrome *c*), potentially involved in electron transport within the respiratory chain and D172_RS04550 (*yfiA*), implicated in the regulation of translation efficiency. Additionally, several hypothetical proteins were identified, representing unexplored functional pathways (Supplementary Data 5). Some of these hypothetical proteins could be linked to previously established roles of the Csr system, such as the regulation of T6SS-related genes. For instance, a previously studied protein, WP_033023891, which is a potential T6SS effector, was shown to directly bind to CsrA *in vitro* (Fig. 6d). These findings highlight the unique targets of the Csr system in *P. fuliginea* BSW20308 compared to *E. coli*, underscoring species-specific regulatory mechanisms and adaptations to its environmental niche.

In general: Is the complete lack of CsrA antagonist RNAs a realistic physiological scenario? Is it possible that in the complete absence of Pf1-3 there is more (and unspecific) binding of CsrA?

Response:

Thank you for your insightful question.

Indeed, in some CsrA-containing bacteria, such as *Bacillus subtilis*, *Campylobacter jejuni*, and others, there are no known sRNAs that antagonize CsrA [Bogacz M, et al. Binding of *Campylobacter jejuni* FliW Adjacent to the CsrA RNA-Binding Pockets Modulates CsrA Regulatory Activity. 11, (2021); Mukherjee S, Yakhnin H, Kysela D, Sokoloski J, Babitzke P, Kearns DB. CsrA–FliW interaction governs flagellin homeostasis and a checkpoint on flagellar morphogenesis in *Bacillus subtilis*. Mol Microbiol 82, 447-461 (2011); Dugar G, et al. The CsrA-FliW network controls polar localization of the dual-function flagellin mRNA in *Campylobacter jejuni*. Nat Commun 7, (2016); Li J, Gulbranson CJ, Bogacz M, Hendrixson DR, Thompson SA. FliW controls growth-phase expression of *Campylobacter jejuni* flagellar and non-flagellar proteins via the post-transcriptional regulator CsrA. 164, 1308-1319 (2018)]. Instead, the regulation of CsrA activity in these organisms is mediated by protein-protein interactions with FliW. However, it is important to note that sRNAs are inherently difficult to detect due to their small size, lack of sequence conservation, and context-specific expression. Therefore, the absence of evidence for sRNAs antagonizing CsrA in these bacteria does not necessarily confirm their complete absence. The presence or

absence of such regulatory RNAs remains unclear and warrants further investigation.

Regarding the complete absence of Pf1-Pf3, one might anticipate that CsrA could exhibit more extensive binding to available RNA targets due to the loss of its primary antagonistic sRNAs. However, this broader binding may also lead to dynamic changes in the levels of other CsrA targets, including sRNAs not previously identified as antagonists or even specific mRNAs that could potentially act as competitive decoys to sequester CsrA. Such compensatory interactions may further complicate the regulatory landscape of CsrA in the absence of Pf1-Pf3. Additionally, the complete removal of Pf sRNAs led to abnormal physiology, as evidenced by inhibited growth, potentially exacerbating the complexity of the situation. Indeed, our RIP-Seq data revealed a progressive reduction in the number of CsrA-binding targets across $\Delta Pf1$, $\Delta Pf12$, and $\Delta Pf123$ mutant strains. Therefore, we think that the complete absence of Pf sRNAs has a complicated impact on the CsrA targets.

We appreciate your insightful question, which has prompted us to include a more detailed discussion of these possibilities in the revised manuscript. By considering the dynamic interplay between CsrA, its known antagonists, and potential compensatory mechanisms, we aim to provide a more comprehensive understanding of CsrA regulation in *Pseudoalteromonas fuliginea*.

We have also outlined the specific changes we made to the manuscript below for your convenience:

Lines 426-430 in the revision. "Consistent with the observed variations in physiology and growth, the loss of varying numbers of Pf sRNAs exerts a significant impact on the CsrA targetomes. Our comparative RIP-seq analysis uncovered numerous distinct targets in the three mutant strains ($\Delta Pf1$, $\Delta Pf12$, and $\Delta Pf123$) compared to the wild-type (WT) strain. Notably, the knockout of Pf sRNAs not only altered the number but also the composition of the CsrA targetomes, indicating a complex interplay rather than a mere quantitative change. Complete absence of Pf sRNAs did not exhibit broader binding to other available RNA targets due to the loss of its primary antagonists. Such a shift could potentially result in changes to the expression levels of other CsrA-regulated genes, including those of previously uncharacterized sRNAs or certain mRNAs that might act as competitive decoys to sequester CsrA."

Paragraph starting line 215: Can the authors provide any somewhat specific phenotypes underlying their observation of potential regulation?

The same goes for the T6SS – are there any phenotypes the authors can provide in the mutants, such as the presence of the corresponding proteins?

Again the same for the biofilms, at least some key players may have been looked into.

Response:

We appreciate your insightful comments regarding potential phenotypes associated with CsrA regulation of cell envelope proteins, T6SS, and biofilm formation. Below, we provide additional details and experimental evidence that address these points and outline our future plans for further exploration:

Cell envelope proteins. Cell envelope proteins often play crucial roles in cellular signaling pathways and exhibit diverse responses to various signals. In our study, we focused on two specific targets within this category, the two-component system regulator

ompR and the Tol/Pal system protein *pal*. Using EMSA assays, we demonstrated that CsrA directly binds to the mRNAs of both *ompR* and *pal*. Additionally, through competitive binding experiments, we validated that the small RNAs Pf1, Pf2, and Pf3 are capable of competitively binding to CsrA in vitro, thereby influencing its association with these mRNA targets. Therefore, we have validated the specific binding of these genes by CsrA. However, not all gene regulation results in observable phenotypes. At least, we didn't observe any significant phenotype changes related to cell envelope proteins. Moreover, exploring potential phenotypes in an environmental strain from the polar ocean is particularly challenging due to the lack of background knowledge about the cell envelope. Therefore, we believe we have done our best by providing additional experiments to validate the targets.

EMSA of 1 μ M CsrA binding to mRNAs of *pal*, *ompR* and *h-ns*, with 1 μ M BSA as the control.

Type VI Secretion System (T6SS). We have validated the interaction of CsrA with mRNAs of three putative T6SS effector proteins (WP_033023891, WP_007377080 and WP_007377877) in vitro and further demonstrated that Pf sRNAs directly participate in competing with CsrA for binding to these mRNAs. Consistent with these findings, our previous proteomic and transcriptomic data revealed expression of T6SS-related genes (Reference #24 in the revised manuscript), including key structural components *hcp* and *vgrG*, as well as effector proteins, in the mutant strains. These observations suggest that Pf sRNAs modulate T6SS activity through CsrA. The same challenges apply to T6SS phenotype characterization. Unlike the model microorganisms such as *E. coli* and some well-studied pathogens, there is a significant lack of background knowledge about the potential target cells of this polar bacterium. Currently, we have no information about the potential cell targets of this polar bacterium, making phenotype characterization impossible at this stage. We believe it is crucial to expand our research on non-model microbes from extreme environments such as the polar regions to better serve the scientific community. This objective drives both our current study and our future endeavors.

Above are EMSA results of CsrA binding to mRNAs of WP_033023891, WP_007377080 and WP_007377877, with BSA as the control.

Biofilm Formation. For biofilm, we have the phenotype data by measuring the biofilm

quantitatively using a crystal violet staining assay. The results indicated that the deletion of Pf sRNAs led to a reduction in biofilm formation. Additionally, EMSA assays confirmed that CsrA binds to biofilm-associated genes *ompR* and *h-ns*, and that Pf sRNAs can competitively interfere with CsrA's binding to these mRNAs. These findings provide strong evidence that the regulatory network involving CsrA and Pf sRNAs impacts biofilm formation.

Figure 7. Influence of Pf sRNAs on biofilm formation across strains. (a) Biofilm formation capabilities across four strains. Quantification of biofilm formation was determined using the crystal violet staining assay (n = 5). Statistically significant differences are indicated by asterisks (****), with P < 0.0001 denoting highly significant differences in biofilm formation between the WT strain and the Pf sRNA mutants. (b) Regulation of genes involved in multiple regulators associated with the biofilm formation by the Csr system in *P. fuliginosa* BSW20308. Squares near gene names show the detection in the WT (left), $\Delta Pf1$ (top), $\Delta Pf12$ (right), and $\Delta Pf123$ (bottom) strains. The dashed line indicates the as-yet-unidentified regulatory effect. Regulators in red are known to play a positive role in biofilm formation, while those in blue denote a negative role. (c) EMSA of 1 μ M CsrA binding to mRNAs of *ompR* (top) and *h-ns* (bottom), with 1 μ M BSA as the control.

In summary, our study has provided initial mechanistic insights into CsrA's regulation of cell envelope proteins, T6SS, and biofilm formation via Pf sRNAs. Currently, we have done our best to further validate these mRNA targets through additional experiments and provided phenotypic characterization when possible. In the future, we are committed to expanding on these findings to promote a deeper understanding of extremophiles from polar regions within the scientific community. We thank the reviewer for raising these valuable points, which have inspired us to extend our investigation into these important aspects in our next endeavors.

Line 320: Hfq, its role and properties need to be explained before.

Response:

Thank you for your insightful comment regarding the need to explain the role and

properties of Hfq. In response, we have revised the relevant section to include a description of Hfq's role as a global RNA chaperone, which stabilizes sRNA-mRNA interactions and facilitates the base-pairing between trans-encoded sRNAs and their targets. Specifically, we have clarified that the interactions between trans-encoded sRNAs and their targets align with Hfq's function, providing a better context for understanding the crosstalk between the Csr and Hfq systems. We hope this revision addresses your concern and improves the clarity of our manuscript. Thank you again for your valuable suggestion.

We have also outlined the specific changes we made to the manuscript below for your convenience:

Lines 40-43 in the revision. sRNAs are known to regulate gene expression post-transcriptionally, often by binding to mRNA or proteins. The regulation of mRNA typically requires the assistance of Hfq, a global RNA chaperone that stabilizes sRNA-mRNA interactions and facilitates their base-pairing. In contrast, sRNAs that regulate proteins often directly interact with the CsrA protein. CsrA antagonistic sRNAs, a key component of the Csr system, usually lack sequence conservation, making their identification by bioinformatics challenging.

Minor

Line 86: between which genes are the Pf2 and Pf3 genes located?

Response:

Thank you for your comment. Pf2 is situated between the genes D172_RS02375 and D172_RS02370, while Pf3 lies between D172_RS16045 and D172_RS16040. We have added the location information of the two sRNAs to the legend of Figure 1, with the details provided in as follows:

Lines 694-697 in the revision. **Figure 1. Discovery and identification of Pf2 and Pf3 sRNAs.** (a) Nucleotide sequences and key features of Pf2 and Pf3 sRNAs. Pf2 is situated between the genes D172_RS02375 and D172_RS02370, while Pf3 lies between D172_RS16045 and D172_RS16040. Conserved upstream sequences of the CsrB family sRNAs are marked in orange. The promoters verified by initiating GFP expression are in blue. The "GGA" motifs are in pink. The rho-independent terminators are in red."

Line 87: Out of how many chromosomes? Omit ,up to'.

Response:

Thank you for your comment. The complete genome of *P. fuliginea* BSW20308 contained two circular chromosomes.

We have added the chromosome information of *P. fuliginea* BSW20308 in line 69 of the resubmitted manuscript. Additionally, we have removed the phrase "up to" in line 93. Thank you again for your careful reading.

Line 69 in the revision. "Previously, we identified the first Csr sRNA, Pf1, in the marine bacterium *P. fuliginea* BSW20308, which possesses two chromosomes and is capable of thriving in the cold Arctic Ocean."

Line 93: maybe add ,predicted' or ,putative' before promoters

Response:

We sincerely thank you for careful reading. We have added 'predicted' before promoters in line 99.

Line 99 in the revision. "Transcriptional fusions of Pf2 and Pf3 promoters with promoter-less GFP (green fluorescence protein) were constructed, and subsequent GFP fluorescence measurements demonstrated that both predicted promoters were functional (Fig. 1c)."

Figure 1c: It would be good if the OD curves could be added to see whether the promoter activity is correlating to specific growth phases.

Response:

We sincerely thank you for this insightful comment, which has helped improve the clarity and comprehensiveness of our study. In response to your suggestion, we have added OD₆₀₀ curves to Figure 1c to illustrate the growth phases corresponding to the promoter activities of Pf2 and Pf3.

(c) Promoter activities of Pf2 and Pf3 during different growth phases. GFP fluorescence intensity (bars) driven by the Pf2 and Pf3 promoters was measured at 6 h, 9 h, 12 h, and 16 h after inoculation, with pRU1701 (empty vector) as a control. The OD₆₀₀ values (pink line) were recorded simultaneously to correlate promoter activities with bacterial growth phases. Data are presented as mean \pm standard deviation (SD) from three biological replicates. Plotted is the mean \pm s.e.m. (***) $P < 0.001$ using Student's t-test).

As shown in the updated Figure 1c (above), the GFP fluorescence intensity driven by the Pf2 and Pf3 promoters was monitored at various time points (6, 9, 12, and 16 hours), and the OD₆₀₀ values were simultaneously recorded. This addition enables a direct comparison between promoter activity and bacterial growth phases. The results demonstrate a strong correlation between promoter activity and growth phases. Specifically, the activity of both Pf2 and Pf3 promoters increased significantly during the exponential growth phase and decreased as the cells entered the stationary phase. These findings provide further evidence of the growth phase-dependent regulation of these sRNAs.

We deeply appreciate your suggestion, as it has significantly enhanced the interpretability of this figure and the manuscript as a whole.

Line 95: add ,heterologously in E. coli'

Response:

We sincerely thank you for careful reading. As suggested by the reviewer, we have added “heterologously in *E. coli*” in line 101:

Line 101 in the revision. “Furthermore, both Pf2 and Pf3 sRNAs were tested for their ability to bind CsrA heterologously in *E. coli* and regulate glycogen synthesis *in vivo*, following previous studies”

Line 108: is it the same or a different genetic context?

Response:

Thank you for your insightful question. Yes, these homologous sequences are all from *Pseudoalteromonas*.

Supplementary Fig.1/Legend: What are the different lanes?

Response:

Thank you for your comment. We apologize for the ambiguity of our description. We have redescribed the legend of Supplementary Fig.1 as follows:

Supplementary Fig. 1. Confirmation of deletion of Pf2 and Pf3 in *P. fuliginea* BSW20308. (a) PCR detection of the Pf2 knockout plasmids conjugation transformation and Pf2 knockout mutants. M, DNA Marker. WT, BSW20308 wild-type strain. Two sets of primers that are specific to the pK18-mobsacB-Ery plasmids (Ery-F/ Ery-R and SacB-F/ SacB-R) were used to verify the successful entry of the Pf2 knockout plasmid into the *P. fuliginea* BSW20308 (lanes 1 and 2). Lanes 3 and 4, the independent colonies that were verified the knockout of Pf2. (b) PCR detection of the Pf3 knockout plasmids conjugation transformation (lanes 1 and 2) and Pf3 knockout mutants (lanes 3 to 8).

We are deeply grateful for your corrections, which have not only enhanced the quality of our paper but also heightened our awareness of the importance of clearly and accurately describing experimental results when writing scientific research papers. We firmly believe that after these revisions, the paper will be more comprehensible and will convey our research findings with greater precision.

Line 155: please mention already here that the FLAG-tagged CsrA is (fully) active. Please show the control.

Response:

Thank you for your valuable comment and for pointing out this important detail. We sincerely apologize for our oversight and appreciate the opportunity to clarify this point. To address your concern, we now explicitly state in the revised manuscript that the 3×FLAG-tagged CsrA is active in the $\Delta Pf1$, $\Delta Pf12$, $\Delta Pf123$, and WT strains. Additionally, we demonstrate that the introduction of the FLAG-tag does not affect the growth of *P. fuliginea* under the examined conditions. These results have been added as Supplementary Fig. 4 and are described in the revised text as follows:

Lines 164-168 in the revision. "Therefore, we constructed 3×FLAG-tagged CsrA in the $\Delta Pf1$, $\Delta Pf12$, $\Delta Pf123$ and WT strains for RIP-Seq analysis. The expression of 3×FLAG-tagged CsrA is active in four strains, and the introduction of the FLAG-tag does not affect the *P. fuliginea* growth under the examined conditions (Supplementary Fig. 4)."

Supplementary Fig. 4. The expression of *P. fuliginea* CsrA and growth curves in WT, $\Delta Pf1$, $\Delta Pf12$ and $\Delta Pf123$. (a) Western blot analysis of CsrA-3xFLAG expression in WT, $\Delta Pf1$, $\Delta Pf12$ and $\Delta Pf123$. Untagged strains were treated as control. (b) Growth curves for *P. fuliginea* BSW20308 and CsrA-3xFLAG tagged strains grow in 2216E medium in duplicate. Error bars are mean \pm s.e.m.

Furthermore, as mentioned in the Materials and Methods section, we ensured the use of appropriate controls for the RIP-seq experiments to validate our findings. Specifically, the blank control (input samples) was not incubated with anti-FLAG M2 paramagnetic beads to mitigate background noise and peak errors. This has now been clarified in the revised manuscript as follows:

Lines 164-168 in the revision. "Therefore, we constructed 3×FLAG-tagged CsrA in the $\Delta Pf1$, $\Delta Pf12$, $\Delta Pf123$ and WT strains for RIP-Seq analysis. The expression of 3×FLAG-tagged CsrA is active in four strains, and the introduction of the FLAG-tag does not affect the *P. fuliginea* growth under the examined conditions (Supplementary Fig. 4). We performed co-immunoprecipitations (coIPs) on late-exponential phase lysates of *csrA*-3×FLAG strains and, as control, their same lysates that have not been incubated with anti-FLAG beads. The conversion of co-purified RNA into cDNA and subsequent deep sequencing yielded an effective mapping of 98.99% of the sequencing reads onto the genome, ranging from 13.68 to 22.33 million reads per library."

We believe these revisions address your concerns thoroughly and improve the clarity and rigor of our manuscript. Thank you again for your careful review and constructive feedback.

Line 181: two-component signal transduction or general signal transduction (or both), please clarify.

Response:

Thank you for your insightful comment. In response, we would like to clarify that in our enrichment analysis results, the term "signal transduction" includes both two-component signal transduction and general signal transduction pathways. For example, genes such as *arcA*, *envZ*, and *ompR* represent two-component signal transduction systems, while genes like *toIC* and *ompA* are associated with general signal transduction. We have revised this description, as detailed in lines 195-206 of the revised manuscript.

Line 213; Materials and Methods: I believe the results, still the strains to be compared should be on the same plate as there are always significant differences between the properties of the soft-agar plates.

Response:

Thank you for raising this concern. We appreciate your attention to experimental details. To address this point, we would like to clarify that the swimming motility assays for the four strains were conducted on the same soft-agar plate to ensure consistency and minimize variability between plates. Under these conditions, we observed a gradual increase in swimming ability among the four strains, following the order of $\Delta Pf123$, $\Delta Pf12$, $\Delta Pf1$, and WT strains. This approach allowed us to draw reliable comparisons across the strains while mitigating potential discrepancies caused by differences in soft-agar plate properties.

It should be noted that although performing the assay on the same plate reduces variability in plate properties, spatial competition between strains on the same plate might influence their motility and limit precise quantification of differences. Nonetheless, we ensured that the experiments were carefully controlled and consistent with our previous findings. The results of the motility assays, which align with our earlier observations, are shown in the figure below.

Supplementary Fig. 4. Motility assay of wild-type (WT) and mutant strains ($\Delta Pf1$, $\Delta Pf12$, and $\Delta Pf123$) on a single soft-agar plate. The motility of each strain was evaluated on a single 2216E plate containing 0.3% agar to minimize inter-plate variability.

We have also added details of this experiment to the *Materials and Methods* section. For your convenience, the revised text is as follows:

Lines 659-661 in the revision. "Strains were cultured to the exponential growth phase (OD_{600} of 0.4), centrifuged, and resuspended in fresh 2216E medium for motility and biofilm assays. Cell motility was measured using semisolid agar plates in triplicate. For each strain, 5 μ l of resuspended cells were placed in the center of a 2216E plate containing 0.3% agar and incubated at 32°C for over 24 h. The colony diameter was measured. To minimize variability between plates, we also performed motility assays on a single 2216E plate containing 0.3% agar, where 5 μ l of resuspended cells from all four strains were inoculated in separate, equidistant positions on the same plate."

Line 278: please add ,static' to the biofilm

Response:

Thank you for your comment. we have added 'static' in line 305:

Line 305 in the revision. "To understand the impact of Pf sRNAs on static biofilm formation of *P. fuliginosa* BSW20308, biofilm was quantitatively measured and compared for the four strains."

Line 301: was detected.

Response:

Thank you for your comment. we have added 'was' in line 344:

Line 344 in the revision. "A decreasing trend of sRNAs **was** detected along with the deletion of one to three Pf sRNAs."

Line 304: the GGA motif

Response:

Thank you for your comment. In accordance with your suggestion, we have revised line 349 of the manuscript to replace 'GGA motif' with 'the GGA motif':

Line 349 in the revision. "Since these putative sRNAs were detected in the CsrA targetomes, the presence of **the** GGA motif was analyzed."

Line 324: dimers

Response:

Thank you for your comment. We have revised line 368 of the manuscript to replace 'dimer' with 'dimers'. We sincerely thank you for careful reading.

Line 368 in the revision. "Both sRNA0652 and sRNA0654 were shown to bind to Hfq hexamer and CsrA **dimers** at different sequence regions (Figs. 8 and 9)."

Reviewer #3 (Remarks to the Author):

The carbon storage regulator (Csr) system is widely distributed across most bacteria and plays a crucial role in regulating various cellular processes. This manuscript reported two newly identified Csr sRNAs, Pf2 and Pf3, in the Arctic bacterium *Pseudoalteromonas fuliginea* BSW20308. The authors further investigated the functional mechanisms of these sRNAs and found that both sRNAs modulate the interactions between the global regulator CsrA and target RNAs, thereby regulating several cellular pathways, including type VI secretion system, cell motility, and cell envelope formation.

Response:

We sincerely thank you for your positive and encouraging feedback on our manuscript. We deeply appreciate the time and effort you invested in reviewing our work and for recognizing the significance of our findings on the functional mechanisms of the *Pf* sRNAs in the Arctic bacterium *Pseudoalteromonas fuliginea*. Your thoughtful comments have been invaluable in helping us refine and improve the manuscript.

We have carefully addressed the specific questions and concerns you raised, providing detailed explanations where clarification was needed. Additionally, we have fully incorporated your constructive suggestions and made the recommended corrections to ensure the manuscript is more precise and comprehensive.

Your insights have greatly contributed to enhancing the clarity and impact of our work, and we are truly grateful for your thoughtful review. Thank you again for your valuable input. The detailed responses to your specific comments are provided below.

1. The authors constructed $\Delta Pf1$, $\Delta Pf12$, $\Delta Pf123$ and WT strains for RIP-Seq analysis. Please provide an explanation for the reason not to construct $\Delta Pf2$, $\Delta Pf3$, and $\Delta Pf23$ strains.

Response:

We are deeply grateful for your meticulous review of our research and the valuable insights you have provided. In response to your questions, we have conducted thorough reflections and compiled the following responses, hoping to address your concerns.

Firstly, the primary objective of this study was to investigate the impact of varying numbers of Csr sRNAs on the CsrA interactome, an area that remains largely unexplored to date. Our research design was grounded in a comprehensive literature analysis and preliminary experimental results. Preliminary studies have indicated that the binding affinities of the three sRNAs to the CsrA protein differ, manifesting in their distinct affinities for CsrA and varying numbers of "GGA motifs" within their sequences. Specifically, Pf1 possesses the highest number of motifs (25), followed by Pf2 (21), and then Pf3 (9). Consequently, we constructed $\Delta Pf1$ as the single sRNA mutation, and in our dual sRNA mutation construction, we chose to first explore the impact of the simultaneous deletion of Pf1 and Pf2 on the CsrA interactome, aiming to gain deeper and more specific mechanistic insights.

Secondly, from an experimental design perspective, we sought to maximize information acquisition under limited experimental conditions. Due to resource and time constraints, it was not feasible for us to conduct exhaustive mutation studies on all possible sRNA combinations. Therefore, based on existing knowledge and experimental feasibility, we selected the combinations most likely to reveal new findings and be most critical for understanding the regulatory mechanisms of CsrA.

To further supplement our study and address your specific question, we have conducted additional experiments, including the construction of $\Delta Pf2$ and $\Delta Pf3$ single-sRNA mutant strains. Growth curve analysis of these mutant strains revealed no significant growth defects compared to the wild type, indicating that the deletion of Pf2 or Pf3 alone does not significantly impact cellular growth under the tested conditions.

Lastly, we are also aware that selecting specific sRNA combinations for study may introduce biases or limitations. Therefore, in future research, we plan to further expand the scope of our experiments, including mutation studies on other sRNA combinations (such as Pf1 and Pf3, Pf2 and Pf3, etc.), to more comprehensively reveal the regulatory network of CsrA and the specific roles of different sRNAs within it.

In summary, our decision to construct $\Delta Pf1$, $\Delta Pf12$, and $\Delta Pf123$ was based on a comprehensive consideration of existing knowledge, the feasibility of experimental design, and planning for future research directions. The additional experiments on $\Delta Pf2$ and $\Delta Pf3$ further validate the compensatory mechanisms among the sRNAs, enriching our understanding of the regulatory interplay within the Csr system. We anticipate that this study will provide new perspectives and clues for understanding the regulatory mechanisms of CsrA and offer valuable references for you and other researchers. Thank you again for your valuable feedback, which has significantly strengthened our manuscript.

2. In Fig. 10, the authors showed two types of sRNAs, "sponge sRNAs" and "base-pairing sRNAs", which are colored in red and green, respectively. Please add this color coding to the figure caption. Additionally, in the right upper panel, which shows the involved cellular pathways, please clarify the square and round shapes represent?

Response:

Thank you for your review. We are very sorry for our negligence of Figure 10. More description on the Figure 10 has been added in the revised manuscript and shown below as well. We sincerely thank you for careful reading. In the right upper panel of Figure 10, circles (round shapes) denote genes that exhibit consistent binding to CsrA protein in WT and ΔPf mutants, and diamonds (square shapes) indicate genes with variable CsrA binding across four strains. We have also highlighted these modifications in the revised manuscript.

Figure 10. Model of crosstalk among CsrA, sRNA, mRNA and Hfq. The CsrA protein functions as a global regulator, binding to mRNA related to the Type VI secretion system, cell motility, and cell envelope etc. Circles denote genes that exhibit consistent binding to CsrA protein in WT and ΔPf mutants. Diamonds indicate genes with variable CsrA binding across the strains. (top right). Meanwhile, CsrA can bind to sRNAs, including red labeled "sponge sRNAs" (e.g., Pf1, Pf2, and Pf3 at top left) and green labeled "base-pairing sRNAs" (bottom). Most trans-encoded sRNAs necessitate the assistance of Hfq protein for

pairing with their targets.